# MBrain: A Multi-channel Self-Supervised Learning Framework for Brain Signals

## Abstract

Brain signals are important quantitative data for understanding physiological activities and diseases of human brain. Meanwhile, rapidly developing deep learning methods offer a wide range of opportunities for better modeling brain signals, which has attracted considerable research efforts recently. Most existing studies pay attention to supervised learning methods, which, however, require high-cost clinical labels. In addition, the huge difference in the clinical patterns of brain signals measured by invasive (*e.g.*, SEEG) and non-invasive (*e.g.*, EEG) methods leads to the lack of a unified method. To handle the above issues, in this paper, we propose to study the self-supervised learning (SSL) framework for brain signals that can be applied to pre-train either SEEG or EEG data. Intuitively, brain signals, generated by the firing of neurons, are transmitted among different connecting structures in human brain. Inspired by this, we propose to learn implicit spatial and temporal correlations between different channels (*i.e.*, contacts of the electrode, corresponding to different brain areas) as the cornerstone for uniformly modeling different types of brain signals. Specifically, we capture the temporal correlation by designing the *delayed-time-shift prediction* task; we represent the spatial correlation by a *graph* structure, which is built with proposed multi-channel CPC whose goal is to maximize the mutual information of each channel and its correlated ones. We further theoretically prove that our design can lead to a better predictive representation and propose the *instantaneou-time-shift prediction* task based on it. Finally, *replace-discriminative-learning* task is designed to preserve the characteristics of each channel. Extensive experiments of seizure detection on both EEG and SEEG large-scale real-world datasets demonstrate our model outperforms several state-of-the-art time series SSL and unsupervised models.

## 1 Introduction

Brain signals are foundational quantitative data for the study of human brain in the field of neuroscience. The patterns of brain signals can greatly help us to understand the normal physiological function of the brain and the mechanism of related diseases. There are many applications of brain signals, such as cognitive research (Ismail & Karwowski, 2020; Kuanar et al., 2018), emotion recognition (Song et al., 2020; Chen et al., 2019), neurological disorders (Alturki et al., 2020; Yuan et al., 2019) and so on. Brain signals can be measured by noninvasive or invasive methods (Paluszek et al., 2015). The noninvasive methods, like *electroencephalography* (EEG), cannot simultaneously consider temporal and spatial resolution along with the deep brain information, but they are easier to implement without any surgery. As for invasive methods like *stereoelectroencephalography* (SEEG), they require extra surgeries to insert the recording devices, but have access to more precise and higher signal-to-noise data. For both EEG and SEEG data, there are multiple *electrodes* with contacts (also called *channels*) that are sampled at a fixed frequency to record brain signals.

Recently, discoveries in the field of neuroscience have inspired advances of deep learning techniques, which in turn promotes neuroscience research. According to the literature, most deep learning-based studies of brain signals focus on supervised learning (Shoeibi et al., 2021; Rasheed et al., 2020; Zhang et al., 2021; Craik et al., 2019), which relies on a large number of clinical labels. However, obtaining accurate and reliable clinical labels requires a high cost. In the meantime, the emergence of self-supervised learning (SSL) and its great success (Chen & He, 2021; Grill et al., 2020; He et al., 2020; Brown et al., 2020; Devlin et al., 2018; Raffel et al., 2020; Van den Oord

et al., 2018) makes it a predominant learning paradigm in the absence of labels. Therefore, some recent studies have introduced the means of SSL to extract the representations of brain signal data. For example, Banville et al. (2021) directly applies general SSL tasks to pre-train EEG data, including relative position prediction (Doersch et al., 2015), temporal shuffling (Misra et al., 2016) and contrastive predictive coding (Van den Oord et al., 2018). Mohsenvand et al. (2020) designs data augmentation methods, and extends the self-supervised model SimCLR (Chen et al., 2020) in computer vision to EEG data. In contrast to the numerous works investigating EEG, few studies focus on SEEG data. Martini et al. (2021) proposes a self-supervised learning model for real-time epilepsy monitoring in multimodal scenarios with SEEG data and video recordings.

Despite the advances on representation learning of brain signals, two main issues remain to be overcome. Firstly, almost all existing methods are designed for a particular type of brain signal data, and there is a lack of a unified method for handling both EEG and SEEG data. The challenge mainly lies in the different clinical patterns of brain signals that need to be measured in different ways. On the one hand, EEG collects noisy and rough brain signals on the scalp; differently, SEEG collects deeper signals with more stereo spatial information, which indicates more significant differences of different brain areas (Perucca et al., 2014). On the other hand, in contrast to EEG with a gold-standard collection location, the monitoring areas of SEEG vary greatly between patients, leading to different number and position of channels. Therefore, how to find the commonalities of EEG and SEEG data to design a unified framework is challenging.

Another issue is about the gap between existing methods and the real-world applications. In clinical scenarios, doctors typically locate brain lesions by analyzing signal patterns of *each* channel and their *holistic* correlations. A straight-forward way for this goal is to model each of the channels separately by single-channel time series models, which, however, cannot exploit correlations between brain areas (Davis et al., 2020; Lynn & Bassett, 2019). As for the existing multivariable time series models, most of them can only capture implicit correlation patterns (Zerveas et al., 2021; Chen & Shi, 2021), whereas explicit correlations are required by doctors for identifying lesions. Moreover, although some graph-based methods have been proposed to explicitly learn correlations, they focus on giving an overall prediction for all channels at a time but overlook the prediction on one specific channel (Zhang et al., 2022; Shang et al., 2021). Therefore, how to explicitly capture the spatial and temporal correlations while giving channel-wise prediction is another issue to be overcome.

In this paper, our main contribution is to propose a multi-channel self-supervised learning framework *MBrain*, which can be generally applied for learning representations of both EEG and SEEG data. In addition, we pay special attention to its application in seizure detection. Based on domain knowledge and data observations, we propose to learn the correlation graph between channels as the common cornerstone for both two types of brain signals. In particular, we employ Contrastive Predictive Coding (CPC) (Van den Oord et al., 2018) as the backbone model of our framework by extending it for handling multi-channel data with theoretically guaranteed effectiveness. Based on the multi-channel CPC, we propose the instantaneous time shift task to learn the spatial correlations between channels, and the delayed time shift task and the replace discriminative task are designed to capture the temporal correlation patterns and to preserve the characteristics of each channel respectively. Extensive experiments show that *MBrain* outperforms several state-of-the-art baselines on large-scale real-world EEG and SEEG datasets for the seizure detection task.

## 2    PRELIMINARY: THEORETICAL ANALYSIS OF MULTI-CHANNEL CPC

We employ Contrastive Predictive Coding (CPC) (Van den Oord et al., 2018) as the basis of our framework. The pretext task of CPC is to predict low-level local representations by high-level global contextual representations $c_t$ at the $t$-th time step. Theoretically, the optimal InfoNCE loss proposed by CPC with $N - 1$ negative samples $\mathcal{L}_N^{\text{opt}}$ is a lower bound of the mutual information between contextual semantic distribution $p(c_t)$ and raw data distribution $p(x_{t+k})$, *i.e.*, $\mathcal{L}_N^{\text{opt}} \geq -I(x_{t+k}; c_t) + \log N$, where $k$ is the prediction step size. CPC is originally designed for single-channel sequence data only, and there are two natural ways to extend single channel CPC to multi-channel version. The first one is to use CNNs with multiple kernels to encode all channels simultaneously, which cannot offer explicit correlation patterns for doctors to identify lesions. The second one is to train a shared CPC regarding all channels as one, which has no ability to capture the correlation patterns. Taking a comprehensive consideration, we propose *multi-channel CPC* in this paper. Our motivation is to explicitly aggregate the semantic information of multiple channels

to predict the local representations of one channel. Formally, we propose the following proposition as our basic starting point.

**Proposition 1.** *Introducing the contextual information of the correlated channels increases the amount of mutual information with the raw data of the target channel.*

$$I(x_{t+k}^i; \Phi(c_t)) = I(x_{t+k}^i; c_t^i, \Phi(\{c_t^j\}_{j \neq i})) \geq I(x_{t+k}^i; c_t^i), \tag{1}$$

*where $i$ and $j$ are indexes of the channels. $\Phi(\cdot)$ represents some kinds of aggregate function, which has no additional formal constraints other than the need to retain information of the target channel.*

*Proof.* We use the linear operation of mutual information to obtain: $I(x_{t+k}^i; c_t^i, \Phi(\{c_t^j\}_{j \neq i})) = I(x_{t+k}^i; c_t^i) + I(x_{t+k}^i; \Phi(\{c_t^j\}_{j \neq i})|c_t^i)$. According to the non-negativity of the conditional mutual information, we complete the proof. □

It seems natural that the predictive ability of multiple channels is stronger than that of a single channel, which is also consistent with the assumption of Granger causality (Granger, 1969) to some extent. Therefore, we choose to approximate the more informative $I(x_{t+k}^i; \Phi(c_t))$ to obtain more expressive representations. Specifically, followed by InfoNCE, we define our loss function $\mathcal{L}_N$ as

$$\mathcal{L}_N = -\sum_i \mathbb{E}_{X^i} \left[ \log \frac{f_k(x_{t+k}, \Phi(c_t))}{\sum_{x_j \in X} f_k(x_j, \Phi(c_t))} \right], \tag{2}$$

where $X^i$ denotes the data sample set consisting of one positive sample and $N - 1$ negative samples of the $i$-th channel. We then establish the relationship between $\mathcal{L}_N$ and $I(x_{t+k}^i; \Phi(c_t))$.

**Theorem 1.** *Given a sample set for each channel $X^i = \{x_1^i, \ldots, x_N^i\}, i = 1, \ldots, n$ consisting of one positive sample from $p(x_{t+k}^i|\Phi(c_t))$ and $N - 1$ negative samples from $\sum_j p(x_{t+k}^j)/n$, where $n$ is the number of channels. The optimal $\mathcal{L}_N^{opt}$ is the lower bound of $\sum_i I(x_{t+k}^i; \Phi(c_t))$:*

$$\mathcal{L}_N^{opt} \geq \sum_i \left[ -I(x_{t+k}^i; \Phi(c_t)) + \log N \right]. \tag{3}$$

*Proof.* The optimal $f_k(x_{t+k}, \Phi(c_t))$ is proportional to $p(x_{t+k}^i|\Phi(c_t))/(\sum_j p(x_{t+k}^j)/n)$, which is the same as single-channel CPC. And we can directly replace the data distributions in the proof of single-channel CPC (see details in Appendix B) to obtain the inequality below:

$$\mathcal{L}_N^{opt} \geq \sum_i \left[ \mathbb{E}_{X^i} \log \left[ \frac{\frac{1}{n} \sum_j p(x_{t+k}^j)}{p(x_{t+k}^i|\Phi(c_t))} \right] + \log N \right] = \mathbb{E}_{X^1, X^2, \ldots, X^n} \log \left[ \frac{[\frac{1}{n} \sum_j p(x_{t+k}^j)]^n}{\Pi_j p(x_{t+k}^j|\Phi(c_t))} \right] + n \log N. \tag{4}$$

According to the Jensen Inequality, we obtain that $(\sum_j \log p(x_{t+k}^j))/n \leq \log(\sum_j p(x_{t+k}^j)/n)$. By exponentiating the two equations, we have

$$\Pi_j p(x_{t+k}^j) \leq [\frac{1}{n} \sum_j p(x_{t+k}^j)]^n. \tag{5}$$

With the help of equation 5, we can further obtain the lower bound of equation 4:

$$\mathcal{L}_N^{opt} \geq \mathbb{E}_{X^1, X^2, \ldots, X^n} \log \left[ \frac{\Pi_j p(x_{t+k}^j)}{\Pi_j p(x_{t+k}^j|\Phi(c_t))} \right] + n \log N = \sum_i \left[ -I(x_{t+k}^i; \Phi(c_t)) + \log N \right]. \tag{6}$$

Then we complete the proof. □

We next analyze the advantages of multi-channel CPC over single-channel CPC. Our loss function $\mathcal{L}_N$ leads to a better predictive representation because we approximate a more informative objective $I(x_{t+k}^i; \Phi(c_t))$, if the optimal loss function for each channel has $\log N$ gap with $I(x_{t+k}^i; \Phi(c_t))$, which is the same in single-channel CPC. Moreover, with the same GPU memory, the more channels, the smaller the batch size that can be accommodated. But we can randomly sample negative samples across all channels, which increases the diversity of negative samples. However, in order to narrow the approximation gap, equation 5 should be considered. The equality sign in this inequality holds if and only if samples from each channel follows the same distribution. In fact, for many large-scale multi-channel time series data (*e.g.*, brain signal data used in this paper), by normalizing each channel, they all exhibit close normal distributions leading to small gaps in equation 5.

## 3 PROPOSED METHOD

In this section, we introduce the details of our proposed self-supervised learning framework *MBrain*. For the commonality between EEG and SEEG, we are inspired by the synergistic effect of brain function and nerve cells, that is, different connectivity patterns correspond to different brain states (Lynn & Bassett, 2019). In particular, for brain signals, nerve cells will spontaneously generate traveling waves and spread them out (Davis et al., 2020), maintaining some characteristics such as shape during the process. Therefore, the degree of channel similarity implies different propagation patterns of traveling waves, reflecting the differences in connectivity patterns to some extent. Both EEG and SEEG brain signals follow the inherent physiological mechanism. Therefore, we propose to extract the correlation graph structure between channels (brain areas) as the cornerstone of unifying EEG and SEEG data (Section 3.1). Next, we introduce three self-supervised learning tasks to model brain signals in Section 3.2. We propose *instantaneous time shift task* based on multi-channel CPC and *delayed time shift task* to capture the spatial and temporal correlation patterns. Then *Replace discriminative task* is further designed to preserve characteristics of each channel.

**Notations.** For both EEG and SEEG data, there are multiple electrodes with $\mathbf{C}$ channels. We use $X = \{x_l \in \mathbb{R}^{\mathbf{C}}\}_{l=1}^{\mathbf{L}}$ to represent raw time series data with $\mathbf{L}$ time points. $i$ and $j$ denote the index of channels. $Y_{l,i} \in \{0, 1\}$ is the label for the $l$-th time point and $i$-th channel. We use a $\mathbf{W}$-length window with no overlap to obtain the time segments $S = \{s_t\}_{t=1}^{|S|}$ (see details in Appendix A). The label corresponding to the $t$-th time segment and the $i$-th channel is denoted as $Y_{t,i}^s$.

### 3.1 LEARNING CORRELATIONS BETWEEN CHANNELS

As mentioned above, the correlation patterns between different brain areas can help us to distinguish brain activities in downstream tasks to a large extent. Taking the seizure detection task as an example, when seizures occur, more rapid and significant propagation of spike-and-wave discharges will appear (Proix et al., 2018), which greatly enhances the correlation between channels. This phenomenon is also verified by data observations in Appendix C, which supports us to treat correlation graph structure learning as the common cornerstone of our framework. However, correlations between brain regions are difficult to be observed and recorded directly. Therefore, for each time step $t$, our goal is to learn the structure of the correlation graph, whose adjacency matrix is $\mathbf{A}_t$, where nodes in the graph indicate channels and weighted edges denote the correlations between channels.

Considering that the brain is in normal and stable state most of the time, we first define the *coarse-grained* correlation graph as the prior graph for a particular individual as

$$\mathbf{A}^{\text{coarse}}(i, j) = \mathbb{E}_{s_t}[\text{Cosine}(s_{t,i}, s_{t,j})], \tag{7}$$

where the expectation operation averages over all the correlation matrices computed in only one time segment $s_t$, and $\text{Cosine}(\cdot, \cdot)$ denotes the cosine similarity function.

Next, based on $\mathbf{A}^{\text{coarse}}$, for each pair of channels, we further model their *fine-grained* short-term correlation within each time segment. We assume that the fine-grained correlations follow a Gaussian distribution element-wise, whose location parameters are elements of $\mathbf{A}^{\text{coarse}}$ and scale parameters will be learned from the data. By means of the reparameterization trick, the short-term correlation matrix of the $t$-th time segment is sampled from the learned Gaussian distribution:

$$\sigma_t(i, j) = \text{SoftPlus}(\text{MLP}(c_{t,\tau,i}^{\text{self}}, c_{t,\tau,j}^{\text{self}})), \tag{8}$$

$$n_t(i, j) \sim \mathcal{N}(0, 1), \tag{9}$$

$$\mathbf{A}_t^{\text{fine}}(i, j) = \mathbf{A}^{\text{coarse}}(i, j) + \sigma_t(i, j) \times n_t(i, j). \tag{10}$$

$\text{SoftPlus}(\cdot)$ is a commonly used activation function to ensure the learned standard deviation is positive. $c_{t,\tau}^{\text{self}}$ is the contextual representation of raw time segments extracted by encoders (see details in Section 3.2). To remove the spurious correlations caused by low frequency signals and enhance the sparsity, which is a common assumption in neuroscience (Yu et al., 2017), we filter the edges by a threshold-based function to obtain the final correlation graph structure $\mathbf{A}_t$:

$$\mathbf{A}_t(i, j) = \begin{cases} \mathbf{A}_t^{\text{fine}}(i, j), & \mathbf{A}_t^{\text{fine}}(i, j) \geq \theta_1, \\ 0, & \mathbf{A}_t^{\text{fine}}(i, j) < \theta_1. \end{cases} \tag{11}$$

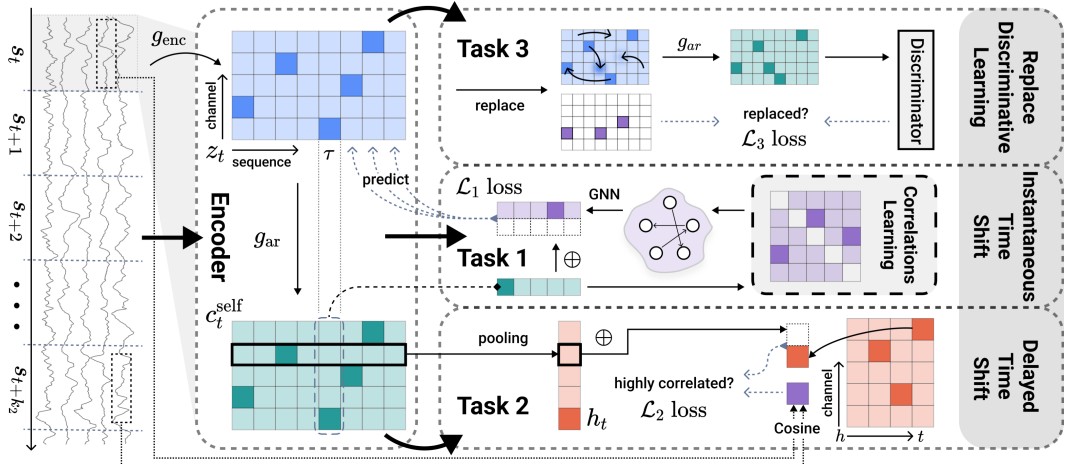

Figure 1: Overview of *MBrain*. The leftmost is the raw multi-channel brain signals. We use an encoder to map the raw data into a low-dimensional representation space. To capture the spatial and temporal correlation patterns, we propose three SSL tasks to guide the encoder to learn informative and distinguishable representations.

## 3.2 SELF-SUPERVISED LEARNING TASKS FOR BRAIN SIGNALS

To capture the correlation patterns in space and time, we propose two self-supervised tasks: *instantaneous time shift* that is based on multi-channel CPC and captures the short-term correlations focusing on spatial patterns; and *delayed time shift* for temporal patterns in broader time scales.

**Instantaneous Time Shift.** For spatial patterns, we aim to leverage the contextual information of correlated channels to better predict future data of the target channel. Therefore, we apply multi-channel CPC and utilize the fine-grained graph structure $\mathbf{A}_t$ obtained in Section 3.1 as the correlations between channels.

We first use a non-linear encoder $g_{enc}$ (1D-CNN with $d$ kernels) mapping the observed time segments to the local latent $d$-dimensional representations $z_t = g_{enc}(s_t) \in \mathbb{R}^{\mathcal{T} \times \mathbf{C} \times d}$ for each channel separately. $\mathcal{T}$ is the sequential length after down sampling by $g_{enc}$. Then an autoregressive model $g_{ar}$ is utilized to summarize the historical $\tau$-length local information of each channel itself to obtain the respective contextual representations:

$$c_{t,\tau}^{self} = g_{ar}(z_{t,1}, \cdots, z_{t,\tau}). \tag{12}$$

In this step, we only extract the contextual information of all channels independently. Based on the graph structure $\mathbf{A}_t$, we instantiate the aggregate function $\Phi(\cdot)$ in equation 4 as GNNs due to their natural message-passing ability on a graph. Here we use a one-layer directed GCN (Yun et al., 2019) to show the process:

$$c_{t,\tau,i}^{other} = \mathrm{ReLU}\left(\frac{\sum_{j \neq i} \mathbf{A}_t(i,j) \cdot c_{t,\tau,j}^{self}}{\sum_{j \neq i} \mathbf{A}_t(i,j)} \cdot \Theta\right), \tag{13}$$

where $\Theta$ is the learnable matrix. Considering that we only aggregate other channels' information, the self-loop in GCN is removed here. Finally, by combining both $c_{t,\tau}^{self}$ and $c_{t,\tau}^{other}$ to obtain the global representations $c_{t,\tau}$, the model can predict the local representations $k_1$-step away $z_{t,\tau+k_1}$ based on the multi-channel CPC loss:

$$c_{t,\tau} = \mathrm{Concat}(c_{t,\tau}^{self}, c_{t,\tau}^{other}), \tag{14}$$

$$\mathcal{L}_1 = \mathcal{L}_N = -\mathbb{E}_{t,i,k_1}\left[\log \frac{c_{t,\tau,i}^{\top} W_{k_1} z_{t,\tau+k_1,i}}{\sum_{z_j \in X_t^i} c_{t,\tau,i}^{\top} W_{k_1} z_j}\right], \tag{15}$$

where $X_t^i$ denotes the random noise set including one positive sample $z_{t,\tau+k_1,i}$ and $N-1$ negative samples. $W_{k_1}$ is the learnable bilinear score matrix of the $k_1$-th step prediction.

**Delayed Time Shift.** For brain areas far apart, there exists delayed brain signal propagation, which is confirmed by the data observations showed in Appendix C. We should consider these significant temporal correlations across several time steps in our model.

Our motivation is that if a simple classifier can easily predict whether two time segments are highly correlated, the segment representations will be significantly different from those with weaker correlations. We thus define the delayed time shift task to encourage more distinguishable segment representations. Similar with instantaneous time shift, we first compute the cosine similarity matrix based on raw data between time segments across several time steps. For the $i$-th channel in the $t$-th time segment, the long-term correlation matrix $\mathbf{B}_t^i$ is computed as

$$\mathbf{B}_t^i(k_2, j) = \text{Cosine}(s_{t,i}, s_{t+k_2,j}), \tag{16}$$

where $j$ traverses all channels including the target channel and $k_2$ traverses at most $K_2$ prediction steps. Then we construct pseudo labels $Y_t^i$ according to $\mathbf{B}_t^i$ to encourage the segment representations with higher correlations to be closer. A predefined threshold $\theta_2$ is set to assign pseudo labels:

$$Y_t^i(k_2, j) = \begin{cases} 1, & \mathbf{B}_t^i(k_2, j) \geq \theta_2, \\ 0, & \mathbf{B}_t^i(k_2, j) < \theta_2. \end{cases} \tag{17}$$

With the pseudo labels, we define the cross entropy loss of the delayed time shift prediction task:

$$h_t = \text{Pooling}(c_{t,1}^{\text{self}}, \cdots, c_{t,\mathcal{T}}^{\text{self}}), \tag{18}$$

$$\hat{p} = \text{Softmax}(\text{MLP}(\text{Concat}(h_{t,i}, h_{t+k_2,j}))), \tag{19}$$

$$\mathcal{L}_2 = -\mathbb{E}_{t,i,k_2,j} \left[ Y_t^i(k_2, j) \log \hat{p} + (1 - Y_t^i(k_2, j)) \log(1 - \hat{p}) \right] \tag{20}$$

where $\hat{p}$ is the predicted probability that the two segments are highly correlated. In practical application, we randomly choose 50% labels from each $Y_t^i$ for efficient training.

**Replace Discriminative Learning.** Consistently exploiting correlation for all channels will weaken the specificity between channels. However, there are significant differences in the physiological signal patterns of different brain areas recorded by channels. Therefore, retaining the characteristics of each channel cannot be ignored for the modeling of brain signals. For this purpose, we further design the replace discriminative learning task.

Following BERT (Devlin et al., 2018), we randomly replace $r\%$ local representations throughout $z_t$ by $\hat{z}_t$, which is sampled from any $\mathcal{T}$ sequences and any $\mathbf{C}$ channels in $z_t$. We use the notation $\mathcal{I}(\hat{z}_t)$ to represent the new local representations after replacement and the corresponding channel indexes of $\hat{z}_t$ in the original sequence. We generate pseudo labels $Y_t$ of the task as below:

$$Y_t(\tau, i) = \begin{cases} 1, & \mathcal{I}(\hat{z}_{t,\tau,i}) \neq i, \\ 0, & \mathcal{I}(\hat{z}_{t,\tau,i}) = i. \end{cases} \tag{21}$$

$\tau$ and $i$ traverse $\mathcal{T}$ sequences and $\mathbf{C}$ channels of $\hat{z}_t$. After obtaining $\hat{z}_t$, we put it into the autoregressive model to get the new contextual representations $\hat{c}_t = g_{\text{ar}}(\hat{z}_t)$. Finally, a simple discriminator implemented by an MLP is utilized to classify whether $\hat{c}_t$ are replaced by other channels or not:

$$\mathcal{L}_3 = -\mathbb{E}_{t,\tau,i} \left[ Y_t(\tau, i) \log \hat{q} + (1 - Y_t(\tau, i)) \log(1 - \hat{q}) \right], \tag{22}$$

where $\hat{q}$ is the predicted probability that $\hat{c}_{t,\tau,i}$ is replaced. When the accuracy of discrimination increases, different channel representations output by the autoregressive model are easier to distinguish. Therefore, the task preserves the unique characteristics of each channel.

Combining the multi-task loss functions equation 15, equation 20 and equation 22, we jointly train *MBrain* with $\mathcal{L} = (1 - \lambda_1 - \lambda_2)\mathcal{L}_1 + \lambda_1\mathcal{L}_2 + \lambda_2\mathcal{L}_3$. After the SSL stage, the segment representations $h_t$ obtained from equation 18 are used for downstream tasks.

## 4 EXPERIMENTS

### 4.1 DATASETS AND BASELINES

**SEEG dataset.** The SEEG dataset used in our experiment is anonymized and provided by a first-class hospital we cooperate with. For a patient suffering from epilepsy, 4 to 10 invasive electrodes

with 52 to 124 channels are used for recording signals. It is worth noting that since SEEG data are collected in a high frequency (1,000Hz or 2,000Hz) through multiple channels, our data is massive. In total, we have collected 470 hours of SEEG signals with a total capacity of 550GB. Professional neurosurgeons help us label the epileptic segments for **each channel**. For the self-supervised learning stage, we randomly sample 5,000 10-second SEEG clips for training and validation. As for the downstream seizure detection task, we first obtain 5,000 sampled 10-second SEEG clips (80% for training and 20% for validation). For the testing, we sample another 2,550 10-second SEEG clips with positive and negative sample ratio of 1:50. We use a 1-second window to segment each clip without overlap and our target is to make predictions for **all channels** in each 1-second segment.

**EEG dataset.** We use the Temple University Hospital EEG Seizure Corpus (TUSZ) v1.5.2 (Shah et al., 2018) as our EEG dataset. It is the largest public EEG seizure database, containing 5,612 EEG recordings, 3,050 annotated seizures from clinical recordings, and eight seizure types. We include 19 EEG channels in the standard 10-20 system. For experimental efficiency, we generate a smaller dataset from TUSZ. We randomly sample 3,000 12-second EEG clips for self-supervised learning. As for the downstream seizure detection task, we first obtain 3,000 sampled 12-second EEG clips (80% for training and 20% for validation). For the testing, we sample another 3,900 12-second EEG clips with positive and negative sample ratio of 1:10. It is worth noting that the labels of EEG data are coarse-grained, which means we only have the label of whether epilepsy occurs in an EEG clip.

**Baselines.** We compare *MBrain* with state-of-the-art models including one supervised time series classification model **MiniRocket** (Dempster et al., 2021) and several self-supervised and unsupervised models: **CPC** (Van den Oord et al., 2018), **SimCLR** (Chen et al., 2020), **Triplet-Loss (T-Loss)** (Franceschi et al., 2019), **Time Series Transformer (TST)** (Zerveas et al., 2021), **GTS** (Shang et al., 2021), **TS-TCC** (Eldele et al., 2021) and **TS2Vec** (Yue et al., 2021). See details of the baselines in Appendix E.

## 4.2 EXPERIMENTAL SETUP

To demonstrate the effectiveness of *MBrain*, we first define the seizure detection task and perform the experiments on EEG and SEEG datasets respectively. To examine the generalization of our model, we further conduct experiments corresponding to two clinically feasible schemes on SEEG dataset. The ablation study, hyperparameter analysis, variance and more experimental results are showed in Appendix G, H and I. We also show the case study of the correlation graph learned by Section 3.1 in Appendix F to confirm the ability of identifying different brain states.

**Task 1** (Seizure Detection). *__Given__ a time-ordered set including $I_\mathcal{S}$ consecutive time segments with the index of the first segment being $t_0$: $\mathcal{S} = \{s_{t_0}, \ldots, s_{t_0+I_\mathcal{S}}\}$, models __predict__ the labels $\hat{Y}_{t,i}^s$ for all time segments in $\mathcal{S}$ (i.e., $t = t_0, \ldots, t_0 + I_\mathcal{S}$) and all channels in each segment (i.e., $i = 1, \ldots, \mathbf{C}$).*

**Seizure detection experiment.** In this experiment, we first perform the self-supervised learning of the model on unlabeled data. Then the segment representations learned by the model are used for downstream seizure detection task (see details in Appendix D). During the training phase of downstream task, the encoder of SSL model will be fine-tuned with a very low learning rate. For EEG data, there is no overlap between the patients of training and testing sets. In addition, since EEG labels are coarse-grained, the segment representations in each 12-second EEG clip are pooled to one representation and then be used for seizure detection. For SEEG data, considering the difference in the number and location of implanted electrodes for patients, we conduct experiments for each patient independently and report the average performance.

**Transfer learning experiments.** To meet practical clinical needs, we design two clinically feasible schemes for SEEG data. **The first is the domain generalization experiment**, that is, training the model on data of existing patients and directly predicting data of unknown patients. More specifically, we follow the "3-1-1" setting, where 3 patients are used for training, 1 patient is used for validation and 1 patient is used for testing. We conduct experiments for all combinations, pick up the best result for each patient, and report the average results over all patients. **The second is the domain adaptation experiment (Motiian et al., 2017).** In this experiment, we first perform SSL on one patient (*i.e.*, source domain) and then fine-tuning is performed using partial labeled data from another patient (*i.e.*, target domain) in the testing set. Finally, we perform seizure detection on the

remaining data of the target domain. We report the results on 12 cross-domain scenarios covering all combinations of four patients with typical seizure patterns in the SEEG dataset.

## 4.3 RESULTS OF SEIZURE DETECTION EXPERIMENT

The average performance of seizure detection on the SEEG dataset are presented in Table 1. Since the positive-negative sample ratio of SEEG dataset is imbalanced, $F$-score is a more appropriate metric to evaluate the performance of models than only considering precision or recall. Especially in clinical applications, doctors pay more attention to finding as much seizures as possible, we thus choose $F_1$-score and $F_2$-score in the experiment. Overall, *MBrain* improves the $F_1$-score by 28.92% and the $F_2$-score by 26.85% on SEEG dataset compared to the best result of baseline methods, demonstrating that *MBrain* can learn informative representations from SEEG data.

Table 1: The average performance of the seizure detection experiment on SEEG and EEG datasets.

| Models | SEEG | | | | EEG | | | | |
|---|---|---|---|---|---|---|---|---|---|
| | Pre. | Rec. | $F_1$ | $F_2$ | Pre. | Rec. | $F_1$ | $F_2$ | AUROC |
| MiniRocket | 22.98 | **66.24** | 31.79 | 43.58 | **22.86** | 63.08 | 33.56 | 46.66 | 75.30 |
| CPC | 27.65 | 55.07 | 34.20 | 42.73 | 22.81 | 58.31 | 32.50 | 44.02 | 74.53 |
| SimCLR | 11.06 | 51.54 | 16.60 | 25.41 | 12.63 | 74.88 | 21.33 | 36.78 | 55.86 |
| T-Loss | 29.29 | 51.55 | 36.00 | 43.13 | 20.72 | 69.25 | 31.82 | 47.00 | 75.88 |
| TST | 13.60 | 44.65 | 19.80 | 28.41 | 15.65 | 28.59 | 19.65 | 23.87 | 58.20 |
| GTS | 24.29 | 40.39 | 29.16 | 34.17 | 18.86 | 62.51 | 28.88 | 42.54 | 71.69 |
| TS-TCC | 22.10 | 49.94 | 25.32 | 32.74 | 15.55 | 39.76 | 21.89 | 29.60 | 58.63 |
| TS2Vec | 30.56 | 52.83 | 36.03 | 43.35 | 21.40 | 58.31 | 31.24 | 43.24 | 73.35 |
| *MBrain* | **37.97** | 65.07 | **46.45** | **55.28** | 22.13 | **76.99** | **34.32** | **51.34** | **77.96** |

Table 1 also shows the results of seizure detection experiment on EEG dataset. Following the common evaluation scheme on EEG dataset (Tang et al., 2022), we also add Area Under the Receiver Operating Characteristic (AUROC) metric in our experiment. Our model is designed to learn the representation for each channel, while there is only one label for an EEG clip. Therefore, it requires the pooling operation to aggregate representations output by our model over channels and time segments for seizure detection. This setting makes the performance improvement of our model not as significant as that in the SEEG experiment. Nevertheless, *MBrain* still outperforms all baselines on $F_1$-score, $F_2$-score and AUROC with an increase of 2.26%, 9.23% and 2.74%, respectively.

Table 2: The average performance of the domain generalization experiment on SEEG dataset.

| Models | Pre. | Rec. | $F_1$ | $F_2$ |
|---|---|---|---|---|
| CPC | 22.88±5.06 | 23.92±**3.90** | 20.11±**3.27** | 21.23±**2.49** |
| T-Loss | 21.38±**4.25** | 28.50±4.07 | 23.48±3.30 | 25.90±3.06 |
| TS2Vec | 27.93±5.23 | 29.49±3.97 | 26.78±3.29 | 27.88±3.52 |
| *MBrain* | **30.69**±5.92 | **38.94**±4.34 | **32.61**±3.60 | **35.64**±3.04 |

## 4.4 RESULTS OF DOMAIN GENERALIZATION EXPERIMENT

In this experiment, we compare the baseline models that perform well in Table 1. This is because, to some extent, the results in Table 1 represent an upper bound on the performance of these models. We point out that although GTS and *MBrain* are both graph-based models, GTS cannot be trained on multiple patients, since the DCRNN (Li et al., 2018) encoder used in GTS can only process graphs with fixed nodes. In contrast, our model is designed to learn the correlations between each pair of nodes and utilizes inductive GNN, so it can easily handle graphs with different numbers of input nodes. In general, the performance of models under the domain generalization setting decreases significantly (40.37% on average in terms of $F_2$-score) compared with that in epilepsy detection

experiment. The drop for recall metric is more pronounced, confirming that the distribution shift of patients in SEEG data is more significant than that in EEG. This results from the fact that different brain regions and different types of epileptic waves have different physiological properties and patterns. *MBrain* in this experiment still improves $F_1$-score and $F_2$-score by 21.77% and 27.83% respectively, compared to the best baseline. The results prove that *MBrain* has a superior generalization ability benefiting from rational inductive assumption of model design.

Table 3: The performance of the domain adaptation experiment on SEEG dataset in terms of $F_2$-score. DA row denotes the performance of *MBrain* in the domain adaptation experiment setting. Max-base and Non-DA rows represent the best performance of baselines and *MBrain* in the seizure detection experiment. Bold numbers and * indicate the best and the second best performance.

| Setting | Group $A$ | | | Group $B$ | | | Group $C$ | | | Group $D$ | | |
|---|---|---|---|---|---|---|---|---|---|---|---|---|
| | $B{\to}A$ | $C{\to}A$ | $D{\to}A$ | $A{\to}B$ | $C{\to}B$ | $D{\to}B$ | $A{\to}C$ | $B{\to}C$ | $D{\to}C$ | $A{\to}D$ | $B{\to}D$ | $C{\to}D$ |
| DA | 68.55 | 69.14* | 68.78 | 41.08 | 46.06 | 46.12* | 40.04 | 39.34 | **48.64** | 80.82* | 79.90 | 80.72 |
| Max-base | | 62.49 | | | 39.78 | | | 33.59 | | | 75.35 | |
| Non-DA | | **70.63** | | | **46.62** | | | 46.09* | | | **83.27** | |

## 4.5 RESULTS OF DOMAIN ADAPTATION EXPERIMENT

According to the results of domain generalization experiment, it is difficult for *MBrain* to achieve competitive performance with Table 1. Due to the long record, clinical SEEG data contains tens or even hundreds of seizures, allowing our model to use a subset of data to fine-tune and then to predict the remaining data. Therefore, as a clinical alternative to domain generalization, we conduct a compromise domain adaptation experiment. Table 3 shows the performance of the domain adaptation (DA) experiment for four patients with typical seizure patterns provided by doctors from SEEG dataset. More specifically, we train *MBrain* on one patient and fine-tune it on all other three patients. $B{\to}A$ denotes that the SSL model is trained on Patient-B and then performs seizure detection experiment with the SSL model being fine-tuned on Patient-A. The results of "Max-base" and "Non-DA" rows correspond to the performance of the best baseline and *MBrain* respectively in scenarios $A{\to}A$, $B{\to}B$, $C{\to}C$ and $D{\to}D$.

Compared with the results in the condition that the self-supervised model and downstream model are both trained on the same patient, the $F_2$-scores of all 12 cross-domain scenarios reduce by less than 15%. Additionally, it can be observed that in all cross-domain scenarios, *MBrain* beats the best baseline in the corresponding scenarios without domain adaptation. It is worth noting that $D{\to}C$ scenario outperforms corresponding "Non-DA" result. The possible reason is that the signal patterns on Patient-D are more significant and recognizable than those on Patient-C. Therefore, the SSL model trained on higher quality source domain can better distinguish signal states when performing downstream tasks on target domain. Overall, the domain adaptation experiment makes *MBrain* achieve competitive performance with Table 1 by fine-tuning it on only a subset of the target domain. The results suggest that *MBrain* captures the inherent features and outputs generalized representations between patients, because we fine-tune the SSL model with a very low learning rate (1e-6). From the perspective of pre-training, the SSL model trained on the source patient gives good initial parameters for the fine-tuning stage on the target patient.

## 5 CONCLUSION AND DISCUSSION

In this paper, we propose a general multi-channel SSL framework *MBrain*, which can be applied for learning representations of both EEG and SEEG brain signals. Based on domain knowledge and data observations, we succeed to use the correlation graph between channels as the cornerstone of our model. The proposed instantaneous and delayed time shift tasks help us capture the correlation patterns of brain signals spatially and temporally. Extensive experiments of seizure detection on large-scale real-world datasets demonstrate the superior performance and generalization ability of *MBrain*. However, there are still some limitations of our work. For example, negative sampling of multi-channel CPC consumes certain memory and time. Besides, we lack a more automatic way to determine the time range of long-term temporal patterns. As for the future work, we plan to collect more types of brain signals and extend *MBrain* to more downstream tasks.

## 6  REPRODUCIBILITY STATEMENT

We provide the source code of our model *MBrain* in the Supplementary Material. Some implementation details of *MBrain* can be found in Appendix D and default hyperparameters can be found in the code. Users can run *MBrain* on their own datasets, following the same generation method of database mentioned in Section 4.1.

## 7  ETHICS STATEMENT

This paper proposes a novel self-supervised learning framework *MBrain* for brain signals, and conduct experiments on real-world large-scale EEG and SEEG datasets. The EEG dataset is public and SEEG dataset is non-public but anonymous. Overall, this work inherits some of the risks of the existing works implementing the EEG dataset and does not introduce any new ethical or future social concerns for SEEG dataset.

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

## A  PRELIMINARIES

**Brain signal data.**  For both EEG and SEEG data, there are multiple *electrodes* with $\mathbf{C}$ contacts that are sampled at a fixed frequency to record the brain signals. We also call these contacts *channels*. For every sampling point, each channel records the potential value of the brain region in which they are located, constituting abstract multi-channel time series data. A complete record file contains a total of $\mathbf{L}$ time points, for which we use the notation $X = \{x_l \in \mathbb{R}^{\mathbf{C}}\}_{l=1}^{\mathbf{L}}$ to represent. In the reminder of this paper, we use $i$ and $j$ to denote the index of channels, such as $x_l = \{x_{l,i}\}_{i=1}^{\mathbf{C}}$. For every $x_{l,i}$, we assign a binary label $Y_{l,i} \in \{0,1\}$ to it according to the start and end time of epileptic brain signals marked by doctors. The time points are in the epileptic state with positive labels ($Y_{l,i} = 1$), while zero labels ($Y_{l,i} = 0$) represent the normal data.

**Preprocessing.**  Following the existing time series works (Zhu, 2017; Bagnall et al., 2017; Schäfer, 2015) with the common preprocessing of segmentation, we use a $\mathbf{W}$-length window to divide the original brain signal data $X$ into time segments $S = \{s_t \in \mathbb{R}^{\mathbf{W} \times \mathbf{C}}\}_{t=1}^{|S|}$ without overlapping. The number of segments $|S| = \lfloor \mathbf{L}/\mathbf{W} \rfloor$. The segment label is obtained from the time points of the whole segment, *i.e.*, $Y_{t,i}^s = \max\{Y_{t \times \mathbf{W}+1,i}, \ldots, Y_{(t+1) \times \mathbf{W},i}\}$.

## B  SINGLE-CHANNEL CPC

Contrastive Predictive Coding (CPC), a pioneering model for self-supervised contrastive learning, sets the pretext task to predict low-level local representations by high-level global contextual information $c_t$. In this way, the model can avoid learning too many details of the raw data and pay more attention to the contextual semantic information of the sequence data. The InfoNCE loss proposed in CPC has become the basic design of the contrastive learning loss function. Specifically, given a raw data sample set $X = \{x_1, \ldots, x_N\}$ consisting of one positive sample from $p(x_{t+k}|c_t)$ and $N-1$ negative samples from the noisy distribution $p(x_{t+k})$, InfoNCE will optimize:

$$\mathcal{L}_N = -\mathbb{E}_X \left[ \log \frac{f_k(x_{t+k}, c_t)}{\sum_{x_j \in X} f_k(x_j, c_t)} \right]. \tag{23}$$

In order to obtain the best classification probability of the positive sample with the cross entropy loss function, the optimal $f_k(x_{t+k}, c_t)$ is proportional to $p(x_{t+k}|c_t)/p(x_{t+k})$. Furthermore, the optimal loss function is also closely related to mutual information, as below:

$$\mathcal{L}_N^{\text{opt}} = -\mathbb{E}_X \left[ \log \frac{p(x_{t+k}|c_t)/p(x_{t+k})}{p(x_{t+k}|c_t)/p(x_{t+k}) + \sum_{x_j \in X_{\text{neg}}} p(x_j|c_t)/p(x_j)} \right]$$

$$\geq \mathbb{E}_X \left[ \log \frac{p(x_{t+k})}{p(x_{t+k}|c_t)} N \right] \tag{24}$$

$$= -I(x_{t+k}; c_t) + \log N. \tag{25}$$

Therefore, we can conclude that while minimizing the loss function $\mathcal{L}_N$, we are also constantly approximating the mutual information of raw data distribution $p(x_{t+k})$ and contextual semantic distribution $p(c_t)$. It turns out that InfoNCE is indeed a well-established loss function designed for self-supervised contrastive learning.

## C  DATA OBSERVATIONS

As Figure 2 shows, for both EEG and SEEG data, we can observe that the correlation matrices are almost the same on two normal segments without overlapping in the same patient. To the opposite, the correlation matrix in the epileptic states differs from the normal ones greatly. These data observations verify the conclusion that correlation patterns can help us to distinguish different brain states, and support us to use the correlation matrix as the cornerstone of EEG and SEEG data.

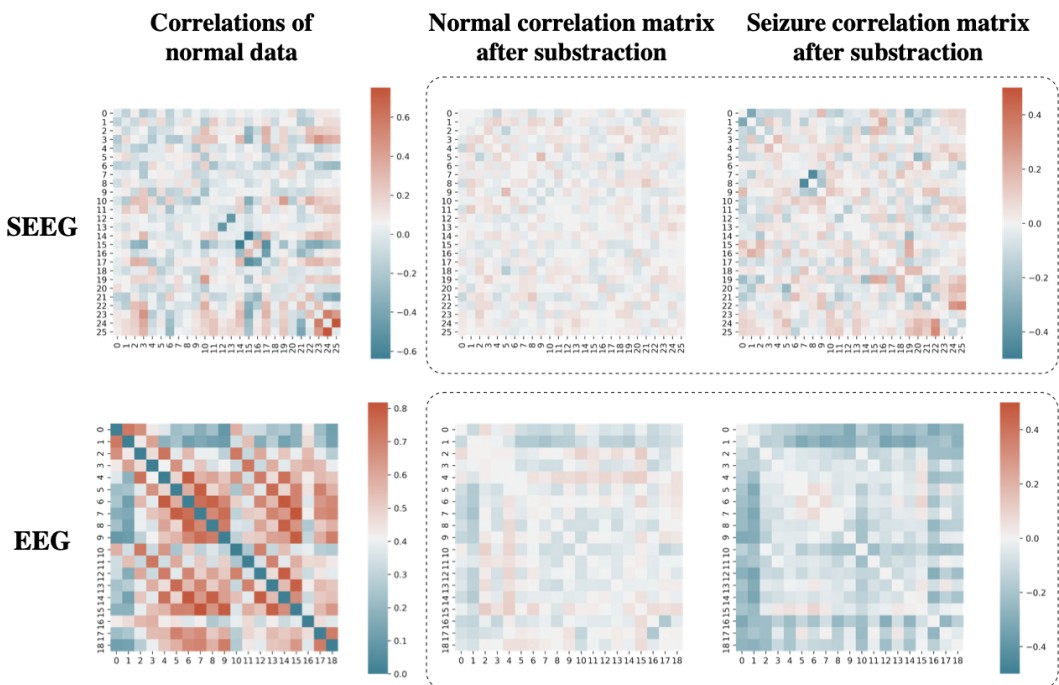

Figure 2: The normal and seizure correlation matrices of EEG and SEEG brain signals. The top row is for SEEG and the bottom row is for EEG. For clear presentation, we sample some channels in SEEG data. The leftmost column including two figures are the base correlation matrices on normal data. The two figures in the middle column represent the matrices after subtracting another normal correlation matrices from the base matrices, and the rightmost column includes matrices after subtracting seizure correlation matrices from the base matrices.

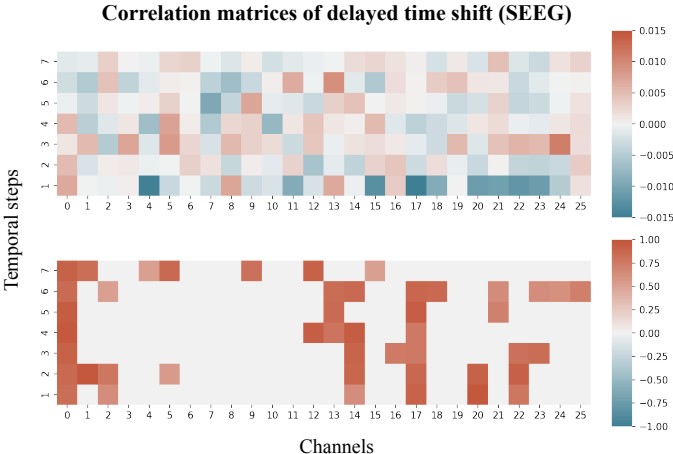

Figure 3: The correlation matrices of delayed time shift of SEEG data. The top figure shows the average correlation matrix over all 10-second SEEG clips of one particular patient. And the bottom figure represents the correlation matrix of one particular sampled 10-second clip. We compute the cosine similarity matrix between the first time segment in the clip of the first channel and the time segments of other channels in the next consecutive 7 time steps. For clear presentation, we sample 26 channels and set correlations below 0.5 to 0 for the bottom figure.

The data observations showed in Figure 3 and Figure 4 confirm that there still exist significant correlations between time segments across several time steps. Unlike instantaneous time shift, delayed correlations are not stable. This can be concluded from the numerical difference between the aver-

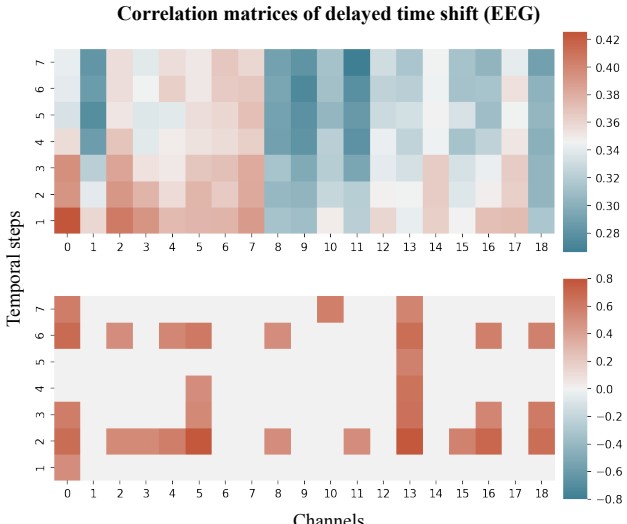

Figure 4: The correlation matrices of delayed time shift of EEG data. The top figure shows the averaged correlation matrix over all 12-second EEG clips. And the bottom figure represents the correlation matrix of one particular 12-second clip which is randomly sampled. The computation and operation are the same as Figure 3.

aged correlation matrix and the one-clip correlation matrix in both figures. Therefore, we design a self-supervised task different from that in the instantaneous time shift to learn the delayed correlations.

## D    IMPLEMENTATION DETAILS OF *MBrain*

The non-linear encoder $g_{\text{enc}}$ used in *MBrain* is composed of three 1-D convolution layers and a one-layer LSTM model (Hochreiter & Schmidhuber, 1997) is used as the autogressive model $g_{\text{ar}}$. The model is optimized using Adam optimizer (Kingma & Ba, 2015) with a learning rate of 2e-4 and weight decay of 1e-6 for the self-supervised learning stage. And for the downstream seizure detection stage, the downstream model is optimized with a learning rate of 5e-4 and weight decay of 1e-6 while the SSL model is fine-tuned with a low learning rate of 1e-6. For the hyperparameters of *MBrain*, we set $\theta_1 = 0.5$ and $\theta_2 = 0.5$. We set the maximum value of $k_1$ in instantaneous time shift task as 8. As Figure 5 shows, we set $K_2 = 7$ so as to take into account the step with the most significant correlation in delayed time shift task. Lastly, we build our model using PyTorch 1.8 (Paszke et al., 2019) and trained it on a workstation with four NVIDIA TESLA T4 GPUs.

For the downstream task, we first utilize an LSTM model (Hochreiter & Schmidhuber, 1997) to encode the segment representations of each channel in chronological order (10-second in SEEG clips and 12-second in EEG clips) independently. One-layer self-attention (Vaswani et al., 2017) is then adopted to all channels within the same time step. Finally, a two-layer MLP classifier is used to predict whether epilepsy is occurring in the time segments. All baselines share the same downstream model in our experiments.

## E    IMPLEMENTATION DETAILS OF BASELINES

- **MiniRocket** (Dempster et al., 2021): Rocket (Dempster et al., 2020) is a state of the art supervised time series classification method based on evaluations on public benchmarks (Bagnall et al., 2017; Tan et al., 2020), involves training a linear classifier on top of features extracted by a flat collection of numerous and various random convolutional kernels. MiniRocket is a variant of Rocket which improves processing time, while offering essentially the same accuracy. We uese the open source

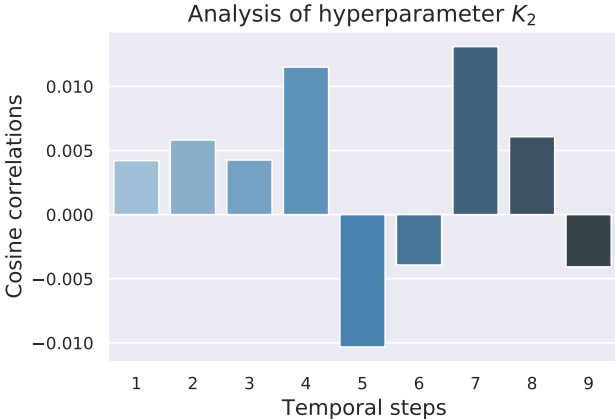

Figure 5: The data observation of how to choose hyperparameter $K_2$. We first average the correlations of all channels of each channel in each time step. Then we average those of all channels in the same time step. As the figure shows, we empirically choose $K_2 = 7$.

code from https://github.com/angus924/minirocket. For each subject, we use the features obtained through MiniRocket to train an independent logistic regression classifier for each channel and test it on the test set of that channel.

- **CPC** (Van den Oord et al., 2018): This is a self-supervised learning method based on a contrastive loss InfoNCE. The pretext task of CPC is set to predict future local low-level representations obtained from multi-layer CNNs by contextual high-level representations obtained from an autoregressive model. This is the backbone model in this paper. We use the open source code of the corrected version from https://github.com/facebookresearch/CPC_audio.

- **SimCLR** (Chen et al., 2020): This is a simple yet effective framework for contrastive learning of visual representations and we use time-series specific augmentations to adapt it to our application. We implemented SimCLR on time series data by ourselves. We use the same encoder architecture and parameter configuration as TS-TCC. In the meantime, we also follow TS-TCC and use scaling (sigma=1.1) as the data augmentation way.

- **Triplet-Loss (T-Loss)** (Franceschi et al., 2019): The approach employs time-based negative sampling and a triplet loss to learn representations for time series segments. We use the default model architecture from the source code provided by the author (https://github.com/White-Link/UnsupervisedScalableRepresentationLearningTimeSeries). For the sampling method of negative samples, we use the data of the previous batch as the candidate set of negative samples of the current batch data (the negative sample candidate set for the first batch is itself). Since the dataloader is shuffled at the end of each epoch, there is no need to worry about the case where the set of sampled negative samples does not change.

- **Time Series Transformer (TST)** (Zerveas et al., 2021): This is a unsupervised representation learning framework for multivariate time series by training a transformer model to extract dense vector representations of time series through an input denoising objective. We use the default model architecture from the source code provided by the author (https://github.com/gzerveas/mvts_transformer).

- **GTS** (Shang et al., 2021): This is a time series forecasting model that learns a graph structure among multiple time series and forecasts them simultaneously with a GNN. In view of this, this model can learn useful representations from unlabeled time series data. We use the default model architecture from the source code provided by the author (https://github.com/chaoshangcs/GTS). In the pre-training stage, we divide each time series segment into 10 parts on average, and learn a time series forecasting model that predicts the next 2 steps based on the previous 8 steps. In the downstream task stage, we use the representation after step 10 as the representation of the time series segment for the seizure detection task.

- **TS-TCC** (Eldele et al., 2021): This is an unsupervised time-series representation learning framework, applying a temporal contrasting module and a contextual contrasting module to learn robust

and discriminative representations. We use the default model architecture from the open source code provided by the author (https://github.com/emadeldeen24/TS-TCC).

- **TS2Vec** (Yue et al., 2021): This is a universal representation learning framework for time series, that applies hierarchical contrasting to learn scale-invariant representations within augmented context views. We use the default model architecture from the source code provided by the author (https://github.com/yuezhihan/ts2vec).

## F CASE STUDY

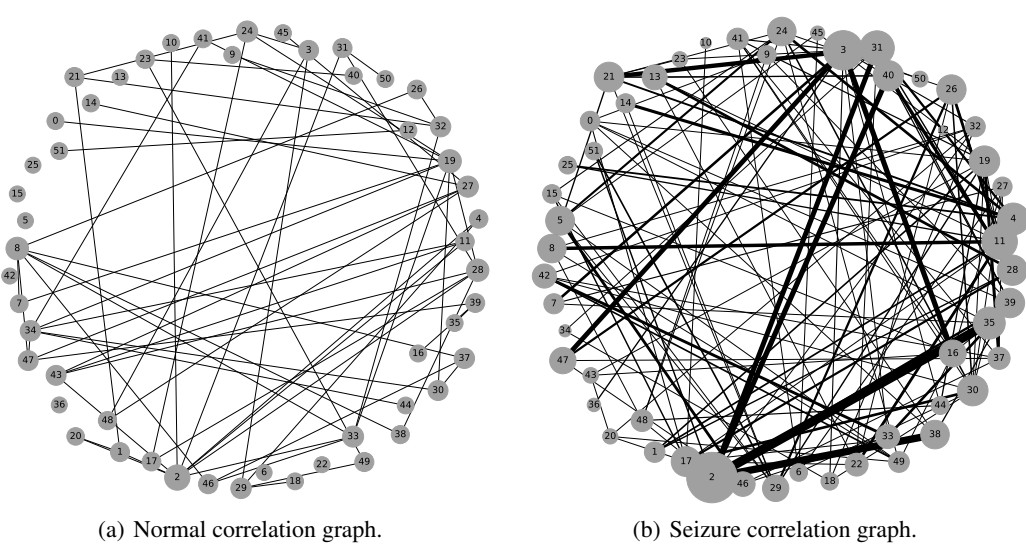

(a) Normal correlation graph.     (b) Seizure correlation graph.

Figure 6: Case study on correlation graphs learned by *MBrain*.

In this section, we study the correlation graphs between the channels learned by *MBrain*. We randomly sample normal and seizure SEEG clips of one particular patient, and visualize their correlation graphs. The correlation graphs showed in Figure 6, we use the threshold $\theta_1$ defined in Section 3.1 for the preservation of the edges. In addition, the thickness of an edge represents the edge weight, and the size of a node represents the sum of the edge weights of all the edges linked to that node. It can be observed that during the normal phase, the correlation is sparser and the weights between edges are smaller, indicating a weaker correlation between channels. In contrast, during the seizure phase, the pattern between channels varies, with the correlation becomes denser and the edge weights become larger. Furthermore, in the correlation graph of the seizure phase, edges with larger weights are usually connected to 2 seizure channels, like Channel-2, Channel-35 and Channel-38 in Figure 6(b), which can help surgeons to better localize the seizure lesions.

## G ABLATION STUDY

We study the effectiveness of each component in *MBrain*. Specifically, we compare *MBrain* with different model variants removing following different components. We firstly remove the correlation graph structure learning module from the instantaneous time shift task and degenerate the task to single-channel CPC while still uniformly sampling negative samples in all channels. This variant is denoted as "*MBrain*-Graph". Next, we respectively remove the whole instantaneous time shift task, the delayed time shift task and replace discriminative task. These model variants are denoted as "*MBrain*-Instant", "*MBrain*-Delay" and "*MBrain*-Replace". Finally, we consider the condition of preserving only one self-supervised task. "*MBrain*-onlyInstant", "*MBrain*-onlyDelay" and

"*MBrain*-onlyReplace" indicate that *MBrain* only performs instantaneous time shift task, delayed time shift task and replace discriminative task respectively.

We repeat the experiments five times in the fine-tuning stage of the downstream seizure detection task. Table 4 shows the results of ablation study on SEEG dataset. It can be observed that the complete *MBrain* achieves the best performance on $F_1$ and $F_2$ metrics, demonstrating the effectiveness of each component in our model design. For "*MBrain*-Instant", the significant decrease in performance illustrates that capturing the spatial and short-term patterns is quite important and is the key to learning the essential representations in multi-channel brain signals. For "*MBrain*-Graph", the decrease in performance demonstrates that multi-channel CPC can greatly help learn more informative representations. Additionally, the performance in "*MBrain*-Delay" and "*MBrain*-Replace" also decreases significantly, illustrating that modeling long-term temporal patterns and preserving the characteristics of channels can help learn more distinguishable representations. In the condition where only one self-supervised task is preserved, it can be observed that the instantaneous time shift task is the most important, which is as expected, and the delayed time shift task and the replace discriminative task contribute similarly to the performance of the complete model.

Table 4: The results of ablation study.

| Models | Pre. | Rec. | $F_1$ | $F_2$ |
|---|---|---|---|---|
| CPC | 27.65±4.49 | 55.07±3.52 | 34.20±3.40 | 42.73±2.57 |
| CPC-Conv | 6.39±**0.77** | 33.21±4.00 | 10.53±**1.07** | 17.46±**1.42** |
| CPC-MLP | 25.84±3.07 | 52.70±3.65 | 32.18±2.46 | 40.34±2.05 |
| *MBrain*-Graph | 36.72±4.59 | 60.48±4.47 | 43.61±3.08 | 51.47±2.68 |
| *MBrain*-Instant | 34.49±4.37 | 55.41±3.90 | 41.57±3.48 | 48.38±2.52 |
| *MBrain*-Delay | 35.00±4.49 | **65.61**±2.94 | 42.97±3.61 | 52.51±1.93 |
| *MBrain*-Replace | 36.08±5.35 | 63.67±4.24 | 43.66±3.66 | 52.49±2.32 |
| *MBrain*-onlyInstant | 36.43±4.44 | 63.66±**2.12** | 43.35±3.83 | 51.82±2.67 |
| *MBrain*-onlyDelay | 31.59±4.24 | 55.03±5.26 | 38.56±2.84 | 46.05±2.26 |
| *MBrain*-onlyReplace | 34.13±6.84 | 56.06±3.68 | 40.02±4.47 | 47.44±2.40 |
| *MBrain* | **37.97**±2.75 | 65.07±2.68 | **46.45**±2.25 | **55.28**±1.77 |

In addition to removing some components and tasks from *MBrain*, we also design some ablation experiments to verify the effectiveness of our proposed graph structure learning. We have proposed two ideas on how to directly implement the multi-channel CPC in Section 2. For the second idea, we have reported the results of a shared CPC regarding all channels as one on the CPC row of Table 1. For the first idea, we design two strategies to combine multi-channel CNN or MLP into CPC respectively to learn representations for each channel.

- Directly use 1-Dimension CNN to encode the whole time series data and the number of channels during the process is $C \rightarrow 256 \rightarrow 256 \rightarrow C \times 256$, and split the output into $C$ representations, each of which is a 256-dimensional representation. Then an LSTM is implemented to it. Then we execute the self-supervised task and the downstream task of CPC based on the representations for each channel as *MBrain* does, this variant is denoted as "CPC-Conv".

- We use the contextual representations of all $n$ channels as input to an MLP in a fixed order, but we set the representation of the target channel to $0$ tensor when we aggregate them. By using the output of MLP as the aggregated representation of other channels, we perform subsequent experiments following exactly the same steps as *MBrain*. We name this variant as "CPC-MLP".

We can observe that the performance of "CPC-Conv" decreases dramatically. We speculate that this is because the channels are relatively independent, and the correlation between most channels is weak or even non-existent. Direct adoption of multi-channel convolution may introduce spurious and noisy correlations. However, the graph structure learning proposed by us has a sparsity assumption, and the representation extraction of each channel is relatively independent, so it can effectively learn and aggregate more significant information. For "CPC-MLP", we use an MLP to aggregate

the representations of other channels, and then concatenate it with the representation of the target channel to predict future data. Unlike "CPC-Con" which directly adopts multi-channel convolution for the raw data to obtain the "mixed" low-level representations, "CPC-MLP", like *MBrain*, learns the correlation of channels based on the "separate" high-level representations. Therefore, the performance of "CPC-MLP" does not drop as dramatically as that of "CPC-Conv". It can be observed that "CPC-MLP" outperforms "CPC" on the EEG dataset. This may result from the fact that the number of channels in EEG dataset is only 19, while that in SEEG dataset is 52 to 124. Consequently, the ablation results show that the graph structure learning we design has a reasonable and parameter-efficient inductive assumption.

## H  HYPERPARAMETER ANALYSIS EXPERIMENTS

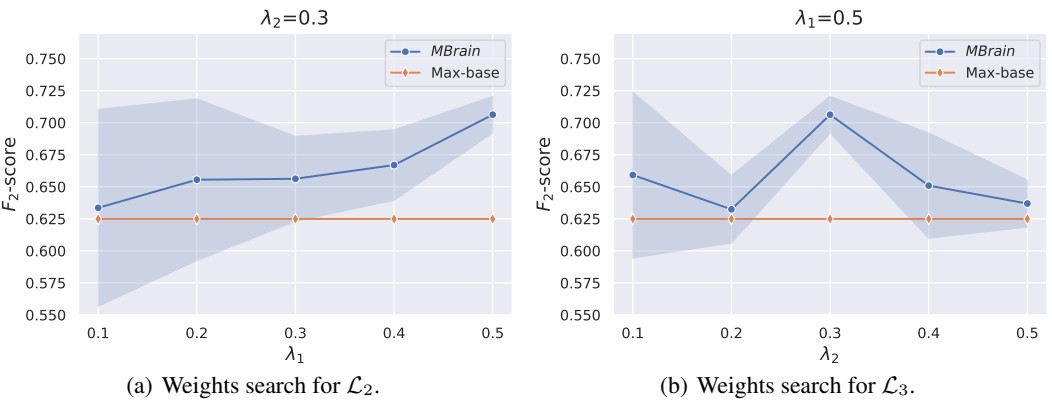

(a) Weights search for $\mathcal{L}_2$.  (b) Weights search for $\mathcal{L}_3$.

Figure 7: Sensitivity analysis on loss weights.

**Sensitivity analysis on loss weights.**  Our loss function is defined as: $\mathcal{L} = (1-\lambda_1-\lambda_2)\mathcal{L}_1 + \lambda_1\mathcal{L}_2 + \lambda_2\mathcal{L}_3$, where $\mathcal{L}_1$, $\mathcal{L}_2$ and $\mathcal{L}_3$ are the loss of instantaneous time shift prediction task, delayed time shift prediction task and replace discriminative task respectively, and $\lambda_1$ and $\lambda_2$ are hyperparameters to balance the three pre-training tasks. We search both of the weights of $\lambda_1$ and $\lambda_2$ in the set $\{0.1, 0.2, 0.3, 0.4, 0.5\}$ and report the tuning results with $F_2$-score for seizure detection task on patient-A from SEEG dataset. In 7(a) and 7(b), we can see that $\lambda_1 = 0.5$ and $\lambda_2 = 0.3$ lead to the optimal performance. In addition, *MBrain* consistently performs better than the best baseline.

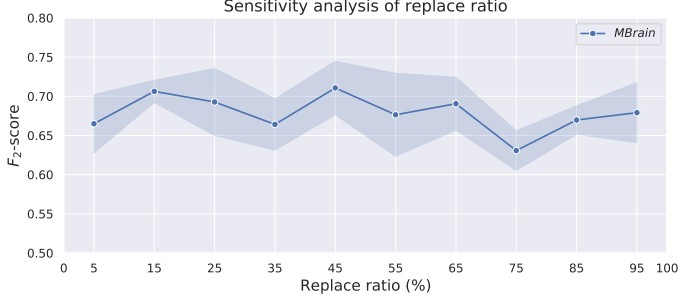

Figure 8: Sensitivity analysis on replace ratio $r\%$.

**Sensitivity analysis on replace ratio.**  We perform sensitivity analysis on replace ratio $r\%$ from replace discriminative task. We search the replace ratio from $5\%$ to $95\%$ and report the tuning results with $F_2$-score for seizure detection task on patient-A from SEEG dataset. As Figure 8 shows, when the replace ratio is set as $45\%$, *MBrain* has the best performance of $71.06\pm3.41$. While *MBrain* gets

the smallest standard deviation and the second best performance of 70.63±1.41 when the replace ratio is set as 15%.

# I  FULL RESULTS

## I.1  FULL RESULTS OF SEIZURE DETECTION EXPERIMENT

Table 5: The average performance of the seizure detection experiment on the SEEG dataset.

| Models | SEEG | | | |
|---|---|---|---|---|
| | Pre. | Rec. | $F_1$ | $F_2$ |
| MiniRocket | 22.98±**0.15** | **66.24**±**0.26** | 31.79±**0.19** | 43.58±**0.22** |
| CPC | 27.65±4.49 | 55.07±3.52 | 34.20±3.40 | 42.73±2.57 |
| SimCLR | 11.06±3.95 | 51.54±5.87 | 16.60±4.68 | 25.41±4.95 |
| T-Loss | 29.29±2.65 | 51.55±2.53 | 36.00±1.97 | 43.13±1.57 |
| TST | 13.60±3.48 | 44.65±4.21 | 19.80±3.73 | 28.41±3.29 |
| GTS | 24.29±4.26 | 40.39±5.80 | 29.16±2.97 | 34.17±2.36 |
| TS-TCC | 22.10±7.65 | 49.94±5.41 | 25.32±8.02 | 32.74±7.95 |
| TS2Vec | 30.56±2.17 | 52.83±2.89 | 36.03±1.72 | 43.35±1.59 |
| *MBrain* | **37.97**±2.75 | 65.07±2.68 | **46.45**±2.25 | **55.28**±1.77 |

Table 6: The average performance of the seizure detection experiment on the SEEG dataset with the encoder of the SSL model frozen.

| Models | SEEG | | | |
|---|---|---|---|---|
| | Pre. | Rec. | $F_1$ | $F_2$ |
| MiniRocket | 22.98±**0.15** | **66.24**±**0.26** | 31.79±**0.19** | 43.58±**0.22** |
| CPC | 26.99±4.35 | 54.16±4.49 | 32.98±3.25 | 41.17±2.45 |
| SimCLR | 10.36±2.76 | 50.89±6.68 | 15.69±3.21 | 24.19±3.34 |
| T-Loss | 29.20±3.21 | 50.24±3.52 | 35.03±1.73 | 41.85±1.99 |
| TST | 13.32±3.96 | 45.59±4.37 | 19.40±3.91 | 28.24±2.86 |
| GTS | 23.13±3.51 | 42.34±4.95 | 28.92±2.70 | 34.78±2.08 |
| TS-TCC | 23.07±7.45 | 50.32±7.11 | 26.29±6.08 | 34.39±5.64 |
| TS2Vec | 30.79±3.08 | 51.43±3.36 | 35.99±2.78 | 42.88±2.44 |
| *MBrain* | **37.72**±5.46 | 63.34±4.27 | **44.47**±4.57 | **53.00**±4.26 |

**Seizure detection experiment.**  First, we keep the initial parameters of the trained self-supervised model unchanged, repeat the experiments five times in the fine-tuning stage of the downstream seizure detection task, and report the mean and standard deviation results in Table 5 and 7. Next, we freeze the entire encoder of SSL model and only train the downstream model and the classifier in the seizure detection task. The same reproducible experimental results are shown in Table 6 and Table 8. We can see that the performance of most models drops slightly compared with the results without freezing the SSL model. Nevertheless, our model still has a competitive performance in $F_2$-score. For the supervised model MiniRocket without a learnable representation extractor, the results remain unchanged in the two experimental settings.

Table 7: The average performance of the seizure detection experiment on the EEG dataset.

| Models | EEG | | | | |
| --- | --- | --- | --- | --- | --- |
| | Pre. | Rec. | $F_1$ | $F_2$ | AUROC |
| MiniRocket | **22.86**±0.84 | 63.08±**1.47** | 33.56±1.11 | 46.66±1.33 | 75.30±0.77 |
| CPC | 22.81±2.04 | 58.31±7.55 | 32.50±1.24 | 44.02±2.43 | 74.53±1.00 |
| SimCLR | 12.63±1.62 | 74.88±16.77 | 21.33±1.95 | 36.78±2.61 | 55.86±5.36 |
| T-Loss | 20.72±1.26 | 69.25±3.99 | 31.82±1.08 | 47.00±**0.50** | 75.88±**0.49** |
| TST | 15.65±1.54 | 28.59±12.93 | 19.65±4.36 | 23.87±8.09 | 58.20±4.27 |
| GTS | 18.86±1.09 | 62.51±5.04 | 28.88±**0.88** | 42.54±1.48 | 71.69±1.88 |
| TS-TCC | 15.55±0.88 | 39.76±11.08 | 21.89±1.20 | 29.60±4.64 | 58.63±1.62 |
| TS2Vec | 21.40±**0.63** | 58.31±6.14 | 31.24±1.18 | 43.24±2.78 | 73.35±1.02 |
| *MBrain* | 22.13±1.03 | **76.99**±4.49 | **34.32**±0.90 | **51.34**±0.97 | **77.96**±0.97 |

Table 8: The average performance of the seizure detection experiment on the EEG dataset with the encoder of the SSL model frozen.

| Models | EEG | | | | |
| --- | --- | --- | --- | --- | --- |
| | Pre. | Rec. | $F_1$ | $F_2$ | AUROC |
| MiniRocket | 22.86±0.84 | 63.08±**1.47** | 33.56±1.11 | 46.66±**1.33** | 75.30±**0.77** |
| CPC | 23.00±1.87 | 53.69±7.47 | 31.90±1.40 | 41.95±3.19 | 73.46±1.35 |
| SimCLR | 11.69±**0.49** | **71.73**±11.90 | 20.05±1.20 | 35.22±3.25 | 51.35±1.16 |
| T-Loss | 21.09±2.61 | 65.54±6.26 | 31.63±2.24 | 45.59±1.51 | 75.25±1.10 |
| TST | 13.42±2.92 | 30.96±13.66 | 17.62±5.05 | 23.11±8.09 | 53.82±5.07 |
| GTS | 18.07±0.78 | 65.01±2.92 | 28.27±**1.07** | 42.76±1.53 | 71.42±2.13 |
| TS-TCC | 14.03±2.50 | 34.34±10.20 | 19.35±3.06 | 25.89±5.68 | 54.80±4.93 |
| TS2Vec | 20.23±1.17 | 64.82±13.92 | 30.43±1.77 | 44.21±5.29 | 73.43±2.60 |
| *MBrain* | **23.09**±1.96 | 66.02±5.05 | **34.08**±1.82 | **47.88**±1.52 | **76.27**±1.91 |

**Significant Analysis.**   Due to the large variance of the experimental results, we further conduct the significance test of the mean values to show that the performance of *MBrain* is indeed significantly better than that of other baseline models. We would like to emphasize that we are primarily concerned with the $F_2$-score of the models, because in the clinical practice, doctors focus on finding as many seizures as possible. Thus, we focus on the significant analysis of $F_2$-score for the results of Table 1.

We perform significance analysis according to the following procedures: (1) For SEEG data, we first combine the repeat results of each model for each patient into one vector. For EEG data, we directly use the vector of repeat results. Significance analysis is performed on F2 score vectors of all models pair by pair. (2) We use Levene's test[1] to test whether two populations have homogeneity of variance, which is a critical property in significant analysis. (3) On the condition of homogeneity of variance, we conduct independent sample T-test[2] for the two populations. We show the p-values of Table 5 and Table 7 in Figure 9 in the form of thermal maps respectively. Considering the symmetry of the significance test, we only show the half of the p-value matrix.

From these two figures, we can see that for EEG data, *MBrain* significantly outperforms other baselines. For SEEG data, although the performance of T-Loss and TS2Vec is not significantly different

---

[1] https://en.wikipedia.org/wiki/Levene's_test
[2] https://en.wikipedia.org/wiki/Student%27s_t-test#Unpaired_and_paired_two-sample_t-tests

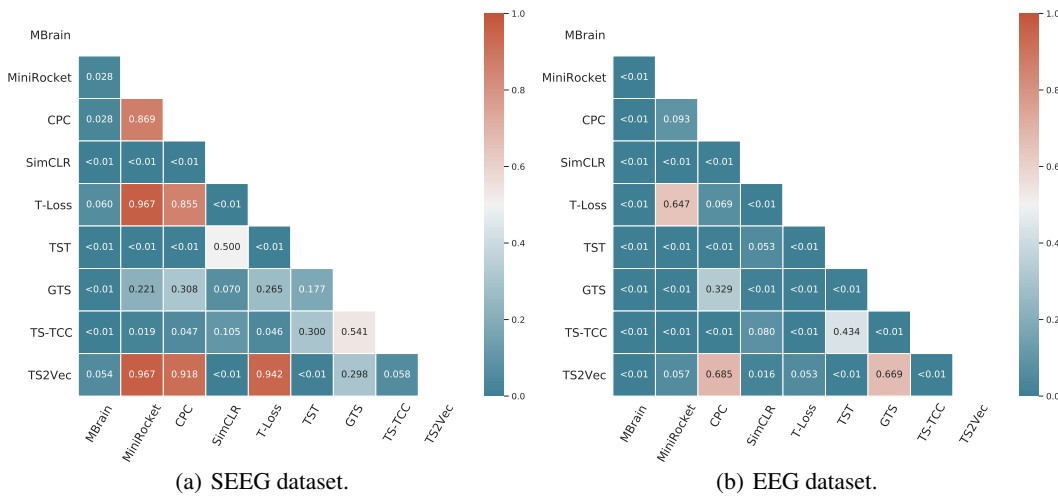

(a) SEEG dataset.  (b) EEG dataset.

Figure 9: Significant analysis.

from that of *MBrain* at the tolerance of 0.05, the p-value was only slightly above 0.05. Therefore, in general, *MBrain* is superior to other baselines models in terms of $F_2$-score. For baselines, we can see that MiniRocket, CPC, T-Loss, TS2Vec and GTS have similar performance on the two datasets. And SimCLR, TST and TS-TCC perform relatively poorly.

## I.2 FULL RESULTS OF DOMAIN ADAPTATION EXPERIMENT

Table 9: The performance of the domain adaptation experiment on SEEG dataset in terms of $F_2$-score. DA row denotes the performance of *MBrain* in the domain adaptation experiment setting. Max-base and Non-DA rows represent the best performance of baselines and *MBrain* in the seizure detection experiment. Bold numbers and * indicate the best and the second best performance.

| Setting | Group $A$ | | | Group $B$ | | |
|---|---|---|---|---|---|---|
| | $B{\to}A$ | $C{\to}A$ | $D{\to}A$ | $A{\to}B$ | $C{\to}B$ | $D{\to}B$ |
| DA | 68.55±4.27 | 69.14±6.54* | 68.78±4.12 | 41.08±2.59 | 46.06±3.05 | 46.12±2.04* |
| Max-base | | 62.49±2.30 | | | 39.78±2.04 | |
| Non-DA | | **70.63±1.41** | | | **46.62±2.42** | |
| Setting | Group $C$ | | | Group $D$ | | |
| | $A{\to}C$ | $B{\to}C$ | $D{\to}C$ | $A{\to}D$ | $B{\to}D$ | $C{\to}D$ |
| DA | 40.04±3.98 | 39.34±2.11 | **48.64±5.48** | 80.82±0.65* | 79.90±1.11 | 80.72±1.31 |
| Max-base | | 33.59±2.23 | | | 75.35±0.79 | |
| Non-DA | | 46.09±2.35* | | | **83.27±0.95** | |

In this experiment, we still keep the self-supervised model trained on the source domain unchanged and repeat the experiments on the target domain five times in the fine-tuning stage of the downstream seizure detection task. The mean and standard deviation results are presented in Table 9. It can be observed that the variances of the results under DA setting are generally higher than those under Non-DA setting, indicating a large variation in the patterns of patients in SEEG dataset. Nevertheless, high-quality source patient data can still be used to pre-train an SSL encoder with good average performance on the target patient. In practice, we can make predictions in an ensemble way by training multiple classifiers.

