# OpenReview forum: "MBrain: A Multi-channel Self-Supervised Learning Framework for Brain Signals"
_ICLR.cc/2023/Conference — Submitted to ICLR 2023_

### Official Review · Reviewer_DmoA · 2022-10-25

**Confidence:** 4
**Correctness:** 3
**Technical Novelty And Significance:** 3
**Empirical Novelty And Significance:** 3
**Recommendation:** 5

**Clarity, Quality, Novelty And Reproducibility:**

#### Clarity
- The paper proposes four components, but the abstract only talks about the delayed time shift and graph module, but in the end, all four components are beneficial. In the introduction, the main contribution is the multi-channel CPC. I found it quite confusing.

#### Quality
- Overall, there is a lack of technical sophistication in the evaluation (statistical significance, proper dataset splits) and empirical evidence in experiments (single vs. multi-channel CPC design). Please refer to Weaknesses for details.

#### Novelty
- The objectives look novel. However, it is hard to evaluate the result's significance without addressing the weaknesses.


**Strength And Weaknesses:**

#### Strength
- The experiments have been performed with SEEG and EEG datasets.
- The authors have considered many baselines.
- The authors have performed an ablation study to show the benefit of each proposed component.

#### Weaknesses
- In section 2 after Theorem 1, it is unclear how the proposed objective is more informative and better. When discussing mutual information for CPC, we talk about lower bounds and tightness. For example, the authors (Tschannen et al., 2019) show that tighter bounds on MI can result in worse representation. While the authors note the gap $\log N$, which is constant and only shifts the lower bound.
- Further, I did not find any experiments with empirical evidence on how multi-channel improves over a single channel without additional modification. The authors provide multiple ideas on how to implement the multi-channel CPC. However, they do not evaluate them. For example, in multimodal learning (Liang et. al, 2022), fusing different modalities with only architectures is expected to improve the performance over unimodal baselines assuming channels are different modalities. So, it could be a simple baseline where the models take multi-channel input, but the loss function is vanilla CPC. I agree with the authors that the correlation is not explicit in this case, but it is still a viable option to improve performance. The comparison could also show the trade-off between performance and interpretability.
- Main experiments have been performed on the dataset splits where the same subject exists in each subset. The authors must evaluate models on the dataset splits that do not overlap on the same subjects. Models can exploit slow features in time series (persistent temporal feature learning) (Feichtenhofer et al., 2021).
- Subject-to-subject domain translation experiments are not stable and depend on the random sample. Consider studying data efficiency over a number of subjects instead of the proposed experiment. Clearly, the model does not generalize in a such low data regime.
- There is no pairwise statistical comparison of the model's performance for significance. Table 1, 2, and 3 does not have a standard deviation, while they are reported in the appendix. Some results, specifically, SimCLR on Recall, have high variance and mean within the variance; thus, selecting the proposed model as bold is misleading. In table 5, CPC is equivalent to the MiniRocket or MBrain.

Feichtenhofer, Christoph, et al. "A large-scale study on unsupervised spatiotemporal representation learning." Proceedings of the IEEE/CVF Conference on Computer Vision and Pattern Recognition. 2021.

Tschannen, Michael, et al. "On mutual information maximization for representation learning." arXiv preprint arXiv:1907.13625 (2019).

Liang, Paul Pu, Amir Zadeh, and Louis-Philippe Morency. "Foundations and Recent Trends in Multimodal Machine Learning: Principles, Challenges, and Open Questions." arXiv preprint arXiv:2209.03430 (2022).

**Summary Of The Paper:**

The manuscript proposes a novel multi-channel self-supervised framework for SEEG and EEG data. The framework comprises delayed-time-shift prediction, instantaneous time shift, replacement discriminative tasks, and an additional graph module. The results show improvements over previous related work and supervised models for multiple performance metrics for seizure detection.

**Summary Of The Review:**

The authors are solving significant problems for healthcare, and there are many exciting ideas: theoretical justification for multi-channel vs. single channel, variants of multi-channel CPC, other self-supervised tasks or modules for EEG and SEEG data, and interpretability (appendix). However, each part of the contribution lacks theoretical justification, technical sophistication, or empirical evidence in a detailed examination; thus, it does not look clear and complete. Overall, It is hard to read the manuscript and understand its significance. Authors should consider exploring one idea at a time and provide concise empirical evidence or theoretical justification for each part.

---

> ### Author Response · Authors · 2022-11-17
> **Response to Weaknesses - Continued**
>
> **Weakness 5 - Continued:**
>
> SEEG data:
>
> | Group 1 | MBrain     | MBrain     | MBrain     | MBrain     | MBrain     | MBrain     | MBrain    | MBrain    | MiniRocket |
> | ------- | ---------- | ---------- | ---------- | ---------- | ---------- | ---------- | --------- | --------- | ---------- |
> | Group 2 | MiniRocket | CPC        | SimCLR     | T-Loss     | TST        | GTS        | TS-TCC    | TS2Vec    | CPC        |
> | p-value | 0.028      | 0.028      | 4.87e-07   | **0.060**  | 1.39e-06   | 0.004      | 3.16e-05  | **0.054** | **0.869**  |
> |         |            |            |            |            |            |            |           |           |            |
> | Group 1 | MiniRocket | MiniRocket | MiniRocket | MiniRocket | MiniRocket | MiniRocket | CPC       | CPC       | CPC        |
> | Group 2 | SimCLR     | T-Loss     | TST        | GTS        | TS-TCC     | TS2Vec     | SimCLR    | T-Loss    | TST        |
> | p-value | 3.58e-04   | **0.967**  | 0.001      | **0.221**  | 0.019      | **0.967**  | 0.001     | **0.855** | 0.005      |
> |         |            |            |            |            |            |            |           |           |            |
> | Group 1 | CPC        | CPC        | CPC        | SimCLR     | SimCLR     | SimCLR     | SimCLR    | SimCLR    | T-Loss     |
> | Group 2 | GTS        | TS-TCC     | TS2Vec     | T-Loss     | TST        | GTS        | TS-TCC    | TS2Vec    | TST        |
> | p-value | **0.308**  | 0.047      | **0.918**  | 0.002      | **0.500**  | **0.070**  | **0.105** | 0.003     | 0.007      |
> |         |            |            |            |            |            |            |           |           |            |
> | Group 1 | T-Loss     | T-Loss     | T-Loss     | TST        | TST        | TST        | GTS       | GTS       | TS-TCC     |
> | Group 2 | GTS        | TS-TCC     | TS2Vec     | GTS        | TS-TCC     | TS2Vec     | TS-TCC    | TS2Vec    | TS2Vec     |
> | p-value | **0.265**  | 0.046      | **0.942**  | **0.177**  | **0.300**  | 0.009      | **0.541** | **0.298** | **0.058**  |
>
> EEG data:
>
> | Group 1 | MBrain     | MBrain     | MBrain     | MBrain     | MBrain     | MBrain     | MBrain    | MBrain    | MiniRocket |
> | ------- | ---------- | ---------- | ---------- | ---------- | ---------- | ---------- | --------- | --------- | ---------- |
> | Group 2 | MiniRocket | CPC        | SimCLR     | T-Loss     | TST        | GTS        | TS-TCC    | TS2Vec    | CPC        |
> | p-value | 0.001      | 0.002      | 1.01e-04   | 1.24e-04   | 0.002      | 4.07e-05   | 5.90e-04  | 0.002     | **0.093**  |
> |         |            |            |            |            |            |            |           |           |            |
> | Group 1 | MiniRocket | MiniRocket | MiniRocket | MiniRocket | MiniRocket | MiniRocket | CPC       | CPC       | CPC        |
> | Group 2 | SimCLR     | T-Loss     | TST        | GTS        | TS-TCC     | TS2Vec     | SimCLR    | T-Loss    | TST        |
> | p-value | 3.37e-04   | **0.647**  | 0.002      | 0.003      | 5.46e-04   | **0.057**  | 0.007     | **0.069** | 0.004      |
> |         |            |            |            |            |            |            |           |           |            |
> | Group 1 | CPC        | CPC        | CPC        | SimCLR     | SimCLR     | SimCLR     | SimCLR    | SimCLR    | T-Loss     |
> | Group 2 | GTS        | TS-TCC     | TS2Vec     | T-Loss     | TST        | GTS        | TS-TCC    | TS2Vec    | TST        |
> | p-value | **0.329**  | 0.002      | **0.685**  | 1.31e-04   | **0.053**  | 0.008      | **0.080** | 0.016     | 0.001      |
> |         |            |            |            |            |            |            |           |           |            |
> | Group 1 | T-Loss     | T-Loss     | T-Loss     | TST        | TST        | TST        | GTS       | GTS       | TS-TCC     |
> | Group 2 | GTS        | TS-TCC     | TS2Vec     | GTS        | TS-TCC     | TS2Vec     | TS-TCC    | TS2Vec    | TS2Vec     |
> | p-value | 4.41e-04   | 3.65e-04   | **0.053**  | 0.005      | **0.434**  | 0.005      | 0.002     | **0.669** | 0.004      |
>
> In these two tables, we bold the items with p-value greater than 0.05, which means that there is no significant difference in the performance of the two compared models under the p-value tolerance of 0.05. We can see that for EEG data, MBrain significantly outperforms other baselines. For SEEG data, although the performance of T-Loss and TS2Vec is not significantly different from that of MBrain at the tolerance of 0.05, the p-value was only slightly above 0.05. Therefore, in general, MBrain is superior to other baselines models in F2 scores. For baselines, we can see that MiniRocket, CPC, T-Loss, TS2Vec and GTS have similar performance on the two datasets. And SimCLR, TST and TS-TCC perform relatively poorly.

---

> > ### Comment · Reviewer_DmoA · 2022-11-20
> > **Response**
> >
> > - Are these results still follow the original split strategy?

---

> ### Author Response · Authors · 2022-11-17
> **Response to Weaknesses - Continued**
>
> **Weakness 4:**
>
> We are sincerely grateful to the suggestion. We have conducted the new domain adaptation experiments with multiple patients as the training set and report the results in **Common Question 1**.
>
> **Weakness 5:**
>
> - We are very sorry that due to the space limitation, we can only put the results of variance in the appendix.
>
> - We are not sure about the problem of misleading bold marks. We would like to know if you mean that we need to compare the mean and variance **separately** and bold them?
>
> - We would like to emphasize that we are primarily concerned with the F2 score of the models, because in the clinical practice, doctors focus on finding as many seizures as possible. Thus, we focus on the significant analysis of F2 score for the results of Table 1.
>   - For SEEG data, we first combine the repeat results of each model for each patient into one vector. For EEG data, we directly use the vector of repeat results. Significance analysis is performed on F2 score vectors of all models pair by pair.
>   - We use [Levene's test](https://en.wikipedia.org/wiki/Levene's_test) to test whether two populations have homogeneity of variance, which is a critical property in significant analysis.
>   - On the condition of homogeneity of variance, we conduct [independent sample T-test](https://en.wikipedia.org/wiki/Student%27s_t-test#Unpaired_and_paired_two-sample_t-tests) for the two populations.
>
>   - We report the p-values as follows:

---

> > ### Comment · Reviewer_DmoA · 2022-11-20
> > **Response**
> >
> > Weakness 5:
> > - I found it reasonable to bold them based on statistical significance.
> > - Have you tested your results on normality before using an independent sample T-test? I suggest using Wilcoxon signed-rank test because it does not require the normality assumption. But for the future, you can also explore ICML 2022 tutorial https://www.cl.uni-heidelberg.de/statnlpgroup/empirical_methods_tutorial/.

---

> ### Author Response · Authors · 2022-11-17
> **Response to Weaknesses**
>
> Thank you very much for your thoughtful feedback and suggestions for improvements. We have addressed your concerns by the following comments.
>
> **Weakness 1:**
>
> - The structure of correlations between the related variables can help predict the target variable. Take two variables ($A$ and $B$) as an example, if the implicit generation rule includes $(a_1,b_1) \rightarrow a_2$ and $(a_1,b_2) \rightarrow a_3$, then only observing $a_1$ cannot accurately predict whether the next value is $a_2$ or $a_3$. Therefore, our proposed objective is more informative in almost all scenarios that include correlated variables. Especially in the brain signal data, the channels are not independent of each other. Channels that are physically close, for example, behave more similarly because they record signals from brain tissues that are closer together and interact with each other.
> - For the optimization goal of multi-channel CPC, we would like to emphasize that by introducing the information of correlated channels, the encoder can benefit from learning the structure and pattern of the correlation between channels, thus obtaining a more informative representation. Moreover, (Tschannen et al., 2019) points out that a proper encoder architecture and a simple scoring function are beneficial to obtain a good representation. Therefore, our model adapting time series encoder architecture empirically proven to show good performance and a simple learnable bilinear matrix as the scoring function, is consistent with the conclusion of (Tschannen et al., 2019) and will not harm the performance of downstream task. More importantly, the conclusions obtained in (Tschannen et al., 2019) are all based on *downstream linear evaluation protocol*. But in practice such as MBrain, a non-linear downstream model is adopted and this regime remains to be further explored, which is also mentioned in conclusion of (Tschannen et al., 2019).
>
> Tschannen, Michael, et al. "On Mutual Information Maximization for Representation Learning." *International Conference on Learning Representations*. 2019.
>
> **Weakness 2:**
>
> Thanks for your suggestion. We report some ablation results in **Common Question 2**.
>
> We would like to emphasize that given **channel-wise** seizure detection on SEEG data, we need to extract a representation and make prediction for **each channel**. And we expect our model to handle patients in SEEG dataset with different channel numbers. These are the main reasons why we come up with a **pair-wise graph structure learning** instead of other designs. For example, our proposed strategy combining CNN and CPC in **Common Question 2** is not transferable between patients because the number of channels varies among SEEG patients. Multimodal strategies are also difficult to apply to our scenario, because the motivation for multimodal learning is to integrate information from different modalities together so that these knowledge can be used together to make predictions. These models are not specifically designed to predict labels for different modalities separately.
>
> **Weakness 3:**
>
> We are sincerely grateful to the suggestion. We report the results of the experiments performed on the SEEG dataset splitting patients as the training and testing sets in **Common Question 1**.
>
> - We agree that models exploit `slow features` when regarding samples within a time range as the positive samples. But we argue that good representations learned from most natural time series data should be inherently smooth in the time domain. Consistent temporal representation learned in the self-supervised learning stage does not necessarily lead to better performance of downstream tasks.
> - The patterns of SEEG signals even from one patient do have differences and variations because of the non-stationarity of SEEG data. We aim to learn the intrinsic generation mechanism/intrinsic distribution behind the patient's brain signals. Training and testing on data from the same patient does not necessarily mean that the model has acquired only shortcuts rather than more general representations. For this reason, the domain adaptation (DA) experiment under the pretrain-finetune paradigm is particularly important.
> - The good performance of our DA experiment exactly suggests that our SSL model captures the common intrinsic mechanism and outputs useful representations between patients, because we fine-tune the SSL model with a very low learning rate (1e-6). From the perspective of pre-training, the SSL model trained on the source patient gives good initial parameters for the fine-tuning stage on the target patient.

---

> > ### Comment · Reviewer_DmoA · 2022-11-20
> > **Response**
> >
> > - Weakness 1: I agree with your idea, but the theoretical part here still requires stricter formulation. I didn't find it general enough, adding more confusion to the paper. The example you used to explain particularly reminds me of a conditional mutual information estimation (Molavipour et al., 2020).
> >   - Molavipour, Sina, Germán Bassi, and Mikael Skoglund. "Conditional mutual information neural estimator." ICASSP 2020-2020 IEEE International Conference on Acoustics, Speech, and Signal Processing (ICASSP). IEEE, 2020.
> >
> > - Weakness 3: I understand your reasoning with SEEG, but SEEG has more degrees of freedom than the same EEG. But we should keep our evaluation strategies the same even there are some data challenges.

---

> > > ### Author Response · Authors · 2022-12-02
> > > **Response**
> > >
> > > - Weakness 1: Do you mean more constraints on the theorem by pointing out `stricter form`? But from the proof, there is no derivation that requires any additional assumptions.
> > > - Weakness 5: Thanks for your suggestion. We have learned a lot and will compare the results with a stricter method.

---

### Official Review · Reviewer_dttW · 2022-10-25

**Confidence:** 4
**Clarity, Quality, Novelty And Reproducibility:** The paper is clearly written. The ide…
**Correctness:** 4
**Technical Novelty And Significance:** 3
**Empirical Novelty And Significance:** 3
**Recommendation:** 8

**Strength And Weaknesses:**

Strengths:
1.	Exploring spatial correlations amongst channels is critical to knowledge discovery from multi-channel brain signal data.
2.	The method is in a self-supervised fashion, which can help alleviate the need for large annotations.
Weaknesses:
1.	Can this method be used on both SEEG and EEG simultaneously?
2.	It would be better to compare with other self-supervised learning methods that are not based on contrastive learning.


**Summary Of The Paper:**

This paper proposes a multi-channel self supervised learning method to overcome two issues in analyzing brain signals: (1) existing methods are often limited to a particular type of brain signal data, either the invasive data (e.g., SEEG) or non-invasive data (e.g., EEG). (2) correlations amongst different brain areas need to be explored to achieve a better understanding of brain activity. To address these issues, the authors propose a CPC-based method on capture the spatial correlations amongst different channels, which also take into account the delayed time shift. Theoretical analysis is also provided. The method is applicable to learning representations of both EEG and SEEG data. The effectiveness has been shown in the application of seizure detection.

**Summary Of The Review:**

Overall, the paper is well written with an interesting idea. Novel self-supervised algorithms have been proposed to explore correlations amongst multiple channels in brain signal data. Effectiveness of the proposed method has been validated using real data from a hospital. It would be more convincing to compare to some non-contrastive-learning-based self-supervised learning methods.

---

> ### Author Response · Authors · 2022-11-17
> **Response to Weaknesses**
>
> We are sincerely grateful to your appreciation of our work.
>
> **Weakness 1:**
>
> Thanks for your suggestion. Technically, our model can process EEG and SEEG data simultaneously. But we take into account that the size of SEEG dataset is smaller than that of EEG, and that they come from completely different sampling devices and different patients. These limitations make it difficult to train the model simultaneously based on the two current datasets. But it is an interesting suggestion, which we will consider in future work.
>
> **Weakness 2:**
>
> Thanks for your suggestion. We design the experiment with a variety of baseline models in mind. Specifically, GTS and TST are both generative models. The difference is that GTS is trained by the prediction loss, while TST is trained by the mask-prediction loss. The rest of the baselines are based on a contrastive loss function. We welcome reviewers to provide us with available baseline models that we have missed but can fit our experimental setting. If feasible, we will conduct the experiments and report the results as soon as possible.

---

> ### Author Response · Authors · 2022-12-04
> **Look forward to your reply**
>
> Dear reviewer dttW, so far, we have revised the article, added experiments or made clarifications through comments according to your suggestions. Do you have any further suggestions? We look forward to your reply and will follow up with your comments promptly.

---

### Official Review · Reviewer_Pj9B · 2022-10-25

**Confidence:** 3
**Correctness:** 3
**Technical Novelty And Significance:** 3
**Empirical Novelty And Significance:** 3
**Recommendation:** 8

**Clarity, Quality, Novelty And Reproducibility:**

- Quality: The connection between the theoretical discussion and the proposed method should be clarified (see above section). But otherwise, seems fine
- Originality: The system is very complex, and for a few design choices, there is not too much justification for the approach taken.The paper could be strengthened by showing that the components of their loss are all necessary, and could not be replaced by simpler components (see above).
- Clarity: Fine. The precise indexing of variables was often helpful.

## small comments/questions
- Proposition 1 proof. What Venn diagram is this referring to? I assume something like this https://en.wikipedia.org/wiki/Information_diagram? Would be good to include it.
- Proposition 1 proof. $\Phi$ needs to be restricted to the set of linear functions, no?
- Equation 2: suggestion -- remove the "*" from the notation.
- Ablation study: "degenerate the task to single-channel CPC" -- does this mean that positive examples are only drawn from a single channel? If so, how is that channel selected?
- Table 1: I know that the results are averaged over subjects, but are they also averaged over channels?
- Table 2: I think I'm missing something. Shouldn't the Non-DA row also have 3 numbers? One for each subject?
- How many patients are used in the SEEG seizure-detection experiment?

**Strength And Weaknesses:**

# Strengths
- Proposes an architecture that can accommodate two very different modalities: EEG and SEEG
- Thorough comparison with other SSL representations
- Domain transfer analysis indicates that this approach can be trained on subjects and then used for held-out subjects

# Weaknesses
- The theoretical discussion in section 2 is a little disconnected from the proposed method, since proposition 1 seems to require that the aggregation function $\Phi$ be linear and non-trivial, but this is not necessarily fulfilled by the learned autoregressive model (eq 12). I believe that in almost all cases, the autoregressive model will not be trivial, but could the authors comment on why they believe the conclusions of proposition 1 should usually hold in practice.
- Section 3.2: There are many ways that multiple channels could be aggregated to form a single context vector. In this work, the context vector is created using a graph, where the edge weights are taken from a correlation matrix. This adds a non-trivial amount of complexity to the system, so it would be appropriate to justify this choice with a few ablation tests. It would be good to compare with aggregation strategies that make fewer assumptions about the structure of the data. For example, a fully connected feed-forward that reduces the dimensionality from $n$ channels to 1 channel could be used instead.
- Section 4.3: I think there is a very simple and informative baseline that should be included: a linear classifier or autoregressive model trained on the same data and targets.
- Section 4.5: An ablation that could be informative: how does the model perform when only the "Replace Discriminative Learning" objective is used? The hyper-parameter analysis shows that this component of the loss has the greatest weight. And similar losses have been sufficient for good performance in the audio domain (see wav2vec2 -- Baevski et al 2020, Mockingjay -- Liu et al 2019).
- Section 4.2. I don't completely understand how domain adaptation is possible between subjects. Doesn't each subject have different electrode placements/a different correlation matrix? How can the pre-trained weights be re-used?

**Summary Of The Paper:**

The authors propose an approach for self-supervised representation learning of brain signal, as recorded by EEG and SEEG. The representations are trained to optimize a loss with three components. The first component is a multi-channel version of the InfoNCE loss, used in contrastive predictive coding. The channels are aggregated according to their instantaneous correlations with each other. The second component gives weight to correlations that may occur at larger time delays. The third component encourages individual channels to be distinguishable. The authors find that their model results in embeddings that can be used downstream for better seizure detection, as compared to other self-supervised representation learning approaches. They further find that these representations can be learned for one subject and then used for prediction on a different subject.

**Summary Of The Review:**

The paper delivers on what it promises: a self supervised representation learning approach for brain signal. Improvements are shown over other reasonable SSL approaches. And transfer learning is shown to be possible. The main downside of this approach is its complexity. The paper could be greatly strengthened if the complexity of each component was justified, especially by way of comparisons to simple linear baselines.

---

> ### Author Response · Authors · 2022-11-14
> **Doubts about a question**
>
> Thank you very much for your questions and suggestions. But we still don't understand one of these problems very well. Would you like to give a more detailed explanation?
>
> - Section 4.3: We are not sure about how to implement the simple baseline. Should we directly conduct supervised training with raw data as input, or participate in fine-tuning as a downstream classifier of self-supervised learning?

---

> > ### Comment · Reviewer_Pj9B · 2022-11-16
> > **Question clarification**
> >
> > I was wondering about the former. Is it possible to do supervised training on a linear classifier with raw data as input?

---

> ### Author Response · Authors · 2022-11-17
> **Response to Small comments/Questions**
>
> **Comment 1:**
>
> We are sorry for the confusion. The Venn diagram is exactly the concept in the link or a more direct link https://en.wikipedia.org/wiki/Venn_diagram. We decide to remove the words because we do not really show how to combine the Venn diagram.
>
> **Comment 2:**
>
> $\Phi$ does not need to be restricted to the set of linear functions. What we want to highlight is extracting information about the target channel and related channels separately before aggregating them, so that we do not lose information about the target channel itself.
>
> **Comment 3:**
>
> Thanks for your suggestion. We will update the notation in the new version.
>
> **Comment 4:**
>
> MBrain-Graph removes the $c^{\text{other}}_{t,\tau}$ in equation (14). The positive samples are still from the target channel which traverses all channels.
>
> **Comment 5:**
>
> To evaluate the results of all channels, we use the `Micro-` like indicators. Specifically, we flatten all channels as one channel to compute the TP, FP, FN and TN. In this way, we are able to avoid the problem that certain indicators cannot be calculated for the channel without a seizure at all. For example, if a channel does not contain any seizures/positive labels, the precision and the denominators of recall and F-score are always 0.
>
> **Comment 6:**
>
> Non-DA shows the specific results in Table 1. Take Group A as an example, we show B/C/D-to-A results from other patients in DA row, and A-to-A result in Non-DA row. More specifically, the Non-DA row shows the results of our model being trained and fine-tuned on the same patient. We want to show that the performance of the domain adaptation experiment can approach or even exceed that of non-transfer experiments, which implies that MBrain has the ability to learn generalized representations between patients.
>
> **Comment 7:**
>
> SEEG data is difficult to obtain, requiring a *craniotomy* to implant electrodes and record data for a long time. The SEEG data requires extensive procedures and approval before it can be used. SEEG dataset currently does not reach the size of EEG and includes five patients in this paper. Although there are only five patients, the record time of each patient is very long (usually more than ten days), and they cover relatively most common types of epilepsy in clinical practice. Therefore, as the initial work of applying AI to epilepsy detection on SEEG, the study based on this dataset still has strong exploratory and practical significance. We will keep working on the problem of generalization between patients based on larger SEEG dataset in future work.

---

> ### Author Response · Authors · 2022-11-17
> **Response to Weaknesses - Continued**
>
> **Weakness 4:**
>
> Thanks for your valuable suggestion. We report the performance of the ablation experiments that preserves only one of the three tasks respectively. "MBrain-onlyInstant", "MBrain-onlyDelay" and "MBrain-onlyReplace" indicate that MBrain only uses instantaneous time shift task, delayed time shift task and replace discriminative task respectively. It can be observed that the instantaneous time shift task is the most important, which is as expected, and the delayed time shift task and the replace discriminative task contribute similarly to the performance of the complete model.
>
> |       Models       |          Pre.           |          Rec.           |          $F_1$          |          $F_2$          |
> | :----------------: | :---------------------: | :---------------------: | :---------------------: | :---------------------: |
> | MBrain-onlyInstant |     $36.43\pm4.44$      | $63.66\mathbf{\pm2.12}$ |     $43.35\pm3.83$      |     $51.82\pm2.67$      |
> |  MBrain-onlyDelay  |     $31.59\pm4.24$      |     $55.03\pm5.26$      |     $38.56\pm2.84$      |     $46.05\pm2.26$      |
> | MBrain-onlyReplace |     $34.13\pm6.84$      |     $56.06\pm3.68$      |     $40.02\pm4.47$      |     $47.44\pm2.40$      |
> |       MBrain       | $\mathbf{37.97\pm2.75}$ | $\mathbf{65.07}\pm2.68$ | $\mathbf{46.45\pm2.25}$ | $\mathbf{55.28\pm1.77}$ |
>
> **Weakness 5:**
>
> Graph structure learning learns the edge weights between **each pair of nodes** (see equation (8)-(10)) instead of the whole graph structure at once, and the directed GCN used in our model is an **[inductive method](https://towardsdatascience.com/inductive-vs-transductive-learning-e608e786f7d)** (see equation (13)). These designs enable the correlation learning module to process graphs with different number of nodes/channels simultaneously.

---

> ### Author Response · Authors · 2022-11-17
> **Response to Weaknesses**
>
> **Weakness 1:**
>
> - The aggregation function $\Phi$ in proposition 1 does not have to be linear. What we want to emphasize here is that in addition to the information of the target channel itself, the introduction of information of related channels can improve the ability to predict the future data of the target channel. We extract the information of target channel and related channels **respectively** and aggregate them (the actual model is also a concatenation operation) to ensure that the representation will not lose the information of the target channel. If we only show the inequality, do you think it will be clearer?
> - In the actual situations, especially in the brain signal data, the channels are not independent of each other. Channels that are physically close, for example, behave more similarly because they record signals from brain tissues that are closer together and interact with each other. The structure of correlations between the related variables can help predict the target variable. Take two variables ($A$ and $B$) as an example, if the implicit generation rule includes $(a_1,b_1) \rightarrow a_2$ and $(a_1,b_2) \rightarrow a_3$, then only observing $a_1$ cannot accurately predict whether the next value is $a_2$ or $a_3$.
>
> **Weakness 2:**
>
> Thanks for your suggestion. We report some ablation results in **Common Question 2**.
>
> The solution of using the full connection feed-forward directly to reduce the dimensionality is a little tricky, because SEEG data usually contains more than 100 channels, which will make the number of parameters much larger relative to graph structure learning. On the other hand, due to the different number of SEEG channels between patients, this solution is difficult to handle the setting of transfer learning. Therefore, we can only conduct the experiment on the same patient.
>
> Even so, the design of the experiment is difficult. We have to make channel-wise prediction on SEEG data, which means that we have to learn a representation for each channel. This situation requires that information from other channels be aggregated for each target channel. If the representations of the remaining $n-1$ channels excluding the target channel are used as input to the MLP,  it will result in misalignment of input representations when information is aggregated for different channels.
>
> To overcome this problem, we still use representations of all $n$ channels as input to the MLP in a fixed order, but we set the representation of the target channel to $0$ tensor when we aggregate them. By using the output of MLP as the aggregated representation of other channels, we perform subsequent experiments following exactly the same steps as MBrain. **Do you think this setting is reasonable?**
>
> **Weakness 3:**
>
> Thanks for your suggestion. We have added two simple and informative baselines as your comments. One is a three-layer MLP and the other is a two-layer bidirectional LSTM. The experimental results of seizure detection on EEG and SEEG datasets are shown in the table below. These two simple baselines cannot achieve competitive performance, probably since the patterns of brain signals are complex and the seizure detection task is also very difficult and non-trivial. Therefore, these two baselines have too few parameters and their model designs are too simple to capture the complex patterns of brain signals well, thus leading to poor performance.
>
> |      | Models |          Pre.           |          Rec.           |          $F_1$          |          $F_2$          |          AUROC          |
> | :--: | :----: | :---------------------: | :---------------------: | :---------------------: | :---------------------: | :---------------------: |
> | SEEG |  MLP   | $7.44\mathbf{\pm0.80}$  | $25.58\mathbf{\pm1.75}$ | $11.22\mathbf{\pm1.08}$ | $16.27\mathbf{\pm1.02}$ |                         |
> |      |  LSTM  |     $13.46\pm2.77$      |     $34.43\pm4.71$      |     $18.44\pm2.40$      |     $24.74\pm1.97$      |                         |
> |      | MBrain | $\mathbf{37.97}\pm2.75$ | $\mathbf{65.07}\pm2.68$ | $\mathbf{46.45}\pm2.25$ | $\mathbf{55.28}\pm1.77$ |                         |
> |      |        |                         |                         |                         |                         |                         |
> | EEG  |  MLP   |     $14.80\pm2.92$      |     $56.96\pm14.51$     |     $22.73\pm1.90$      |     $34.72\pm2.87$      |     $59.26\pm2.51$      |
> |      |  LSTM  |     $16.04\pm1.16$      |     $38.41\pm10.49$     |     $22.07\pm0.99$      |     $29.21\pm4.37$      |     $61.62\pm1.30$      |
> |      | MBrain | $\mathbf{22.13\pm1.03}$ | $\mathbf{76.99\pm4.49}$ | $\mathbf{34.32\pm0.90}$ | $\mathbf{51.34\pm0.97}$ | $\mathbf{77.96\pm0.97}$ |

---

> ### Author Response · Authors · 2022-12-04
> **Look forward to your reply**
>
> Dear reviewer Pj9B, so far, we have revised the article, added experiments or made clarifications through comments according to your suggestions. Do you have any further suggestions? We look forward to your reply and will follow up with your comments promptly.

---

> > ### Comment · Reviewer_Pj9B · 2022-12-06
> > **Increased score**
> >
> > Hello,
> >
> > I thank the authors for their thorough response. In particular, they have added:
> > - Comparisons with a naive aggregation function
> > - Comparison with a simple MLP and LSTM baseline
> > - Ablations for their loss terms
> >
> > These help justify the complexity of their method, which was a concern for me. So I am happy to increase the score.
> >
> > ** small notes
> > - AUROC is missing for the SEEG results in the MLP and LSTM baseline

---

> > > ### Author Response · Authors · 2022-12-07
> > > **Thanks**
> > >
> > > Thank you for raising the score and helping us sort the list of supplementary experiments. We are sincerely grateful to your appreciation of our work. We do not compute AUROC metric for all experiments on SEEG dataset, but we can provide the metric results in the final version. However, since we fail to save all the predicted logits of the repeated experiments, would you accept that we provide the new results of repeated experiments in the final version?

---

### Official Review · Reviewer_9fFf · 2022-10-26

**Confidence:** 3
**Clarity, Quality, Novelty And Reproducibility:** The paper is explained well, especial…
**Correctness:** 4
**Technical Novelty And Significance:** 4
**Empirical Novelty And Significance:** 3
**Recommendation:** 6

**Strength And Weaknesses:**

Strength: SSL on EEG has been studied before, but the authors generalize the idea to SEEG signals as well, and tries to learn correlation graphs which helps with interpretability of the model. Experimental results on seizure detection are encouraging, and are compared to other existing models.

Weakness: Figure 1 is a little confusing. Also, can the authors comment on the large standard deviation values in Tables 5-7? Is the improvement by the proposed method significant given that observation? How long does the fine-tuning take, and what do the prediction results look like without fine-tuning? What happens when the SEEG channels and EEG channels don't overlap?

**Summary Of The Paper:**

This paper proposes a semi-supervised learning (SSL) method that pre-trains on both SEEG and EEG data. It learns the correlation graph between channels from three SSL tasks: instantaneous time shift task, delayed time shift task, and replace discriminative task. Experiments on seizure detection is described with promising results compared to other semi-supervised learning methods.

**Summary Of The Review:**

The idea of using SSL on both EEG and SEEG signals is interesting, and learning correlation graphs improves the interpretability of the model, which can be useful for clinicians.

---

> ### Author Response · Authors · 2022-11-14
> **Doubts about some questions**
>
> Thank you very much for your questions and suggestions. But we still don't understand some of these problems very well. Would you like to give a more detailed explanation?
>
> - Could you please point out exactly what is confusing for Figure 1? We can make targeted modifications according to your comments.
>
> - Could you please describe what the overlap of SEEG channels and EEG channels is more specifically? We are sorry we do not fully understand the question.

---

> ### Author Response · Authors · 2022-11-17
> **Response to Weaknesses - Continued**
>
> - Thanks for your valuable suggestion. For the results of Table 1 in our paper, the encoder of SSL model will be fine-tuned with a very low learning rate during the training phase of downstream task. Now, we froze the encoder of SSL model and only train the classifier in the downstream seizure detection task. The experimental results are shown in the table below. We can see that the performance of most models drops slightly compared with the results without freezing the SSL model. Nevertheless, our model still has a competitive performance in F2 score.
>
> |      |   Models   |          Pre.           |           Rec.           |          $F_1$          |          $F_2$          |          AUROC          |
> | :--: | :--------: | :---------------------: | :----------------------: | :---------------------: | :---------------------: | :---------------------: |
> | SEEG | MiniRocket | $22.98\mathbf{\pm0.15}$ | $\mathbf{66.24\pm0.26}$  | $31.79\mathbf{\pm0.19}$ | $43.58\mathbf{\pm0.22}$ |                         |
> |      |    CPC     |     $26.99\pm4.35$      |      $54.16\pm4.49$      |     $32.98\pm3.25$      |     $41.17\pm2.45$      |                         |
> |      |   SimCLR   |     $10.36\pm2.76$      |      $50.89\pm6.68$      |     $15.69\pm3.21$      |     $24.19\pm3.34$      |                         |
> |      |   T-Loss   |     $29.20\pm3.21$      |      $50.24\pm3.52$      |     $35.03\pm1.73$      |     $41.85\pm1.99$      |                         |
> |      |    TST     |     $13.32\pm3.96$      |      $45.59\pm4.37$      |     $19.40\pm3.91$      |     $28.24\pm2.86$      |                         |
> |      |    GTS     |     $23.13\pm3.51$      |      $42.34\pm4.95$      |     $28.92\pm2.70$      |     $34.78\pm2.08$      |                         |
> |      |   TS-TCC   |     $23.07\pm7.45$      |      $50.32\pm7.11$      |     $26.29\pm6.08$      |     $34.39\pm5.64$      |                         |
> |      |   TS2Vec   |     $30.79\pm3.08$      |      $51.43\pm3.36$      |     $35.99\pm2.78$      |     $42.88\pm2.44$      |                         |
> |      |   MBrain   | $\mathbf{37.72}\pm5.46$ |      $63.34\pm4.27$      | $\mathbf{44.47}\pm4.57$ | $\mathbf{53.00}\pm4.26$ |                         |
> |      |            |                         |                          |                         |                         |                         |
> | EEG  | MiniRocket |     $22.86\pm0.84$      | $63.08\mathbf{\pm1.47}$  |     $33.56\pm1.11$      | $46.66\mathbf{\pm1.33}$ | $75.30\mathbf{\pm0.77}$ |
> |      |    CPC     |     $23.00\pm1.87$      |      $53.69\pm7.47$      |     $31.90\pm1.40$      |     $41.95\pm3.19$      |     $73.46\pm1.35$      |
> |      |   SimCLR   | $11.69\mathbf{\pm0.49}$ | $\mathbf{71.73}\pm11.90$ |     $20.05\pm1.20$      |     $35.22\pm3.25$      |     $51.35\pm1.16$      |
> |      |   T-Loss   |     $21.09\pm2.61$      |      $65.54\pm6.26$      |     $31.63\pm2.24$      |     $45.59\pm1.51$      |     $75.25\pm1.10$      |
> |      |    TST     |     $13.42\pm2.92$      |     $30.96\pm13.66$      |     $17.62\pm5.05$      |     $23.11\pm8.09$      |     $53.82\pm5.07$      |
> |      |    GTS     |     $18.07\pm0.78$      |      $65.01\pm2.92$      | $28.27\mathbf{\pm1.07}$ |     $42.76\pm1.53$      |     $71.42\pm2.13$      |
> |      |   TS-TCC   |     $14.03\pm2.50$      |     $34.34\pm10.20$      |     $19.35\pm3.06$      |     $25.89\pm5.68$      |     $54.80\pm4.93$      |
> |      |   TS2Vec   |     $20.23\pm1.17$      |     $64.82\pm13.92$      |     $30.43\pm1.77$      |     $44.21\pm5.29$      |     $73.43\pm2.60$      |
> |      |   MBrain   | $\mathbf{23.09}\pm1.96$ |      $66.02\pm5.05$      | $\mathbf{34.08}\pm1.82$ | $\mathbf{47.88}\pm1.52$ | $\mathbf{76.27}\pm1.91$ |

---

> ### Author Response · Authors · 2022-11-17
> **Response to Weaknesses**
>
> - We speculate that the variance of the experimental results is related to the nature of the brain signal data and the representations learned by the models. Specifically, for the epilepsy detection task, there are epileptic interictal waves in normal brain signals. These waveforms are similar to some parts of real epileptic seizure waveforms, but we do not distinguish them from normal waves due to the wide distribution and large number of epileptic interictal waves. Therefore, the performance of the models may be affected by them during the supervised learning stage. On the other hand, the variance of different models is also different. The MiniRocket is an almost deterministic model and therefore has relatively stable performance. The large variances of SimCLR, TST and TS-TCC indicate to some extent that the representations learned by these models are unstable. In addition, according to reviewer DmoA's suggestion, we have made a significance analysis. Please refer to our response to **Weakness 5** proposed by reviewer DmoA for the specific results (https://openreview.net/forum?id=ashgrQnYsm&noteId=sq4XjVjHS7).
>
> - Thanks for your valuable suggestion. We report **the averaged time that the models take in the fine-tuning stage**, and the results are shown in the table below. Since we add an early stop mechanism on the validation set to all baselines to prevent models from overfitting, so the number of epochs that each model takes during fine-tuning stage is different.
>
>   | Models | Fine-tuning Time (hours) on SEEG dataset | Fine-tuning Time (hours) on EEG dataset |
>   | :----: | :--------------------------------------: | :-------------------------------------: |
>   |  CPC   |                   0.30                   |                  0.17                   |
>   | SimCLR |                   1.32                   |                  0.08                   |
>   | T-Loss |                   0.22                   |                  0.12                   |
>   |  TST   |                   0.30                   |                  0.17                   |
>   |  GTS   |                   0.65                   |                  1.09                   |
>   | TS-TCC |                   1.41                   |                  0.10                   |
>   | TS2Vec |                   0.60                   |                  0.18                   |
>   | MBrain |                   0.45                   |                  0.34                   |

---

> ### Author Response · Authors · 2022-12-04
> **Look forward to your reply**
>
> Dear reviewer 9fFf, so far, we have revised the article, added experiments or made clarifications through comments according to your suggestions. Do you have any further suggestions? We look forward to your reply and will follow up with your comments promptly.

---

> > ### Comment · Reviewer_9fFf · 2022-12-06
> > **acknowledgement**
> >
> > I'm sorry for the delay. Thank you for the additional results, including the ablation studies and baseline comparisons done for reviewer Pj9B. The revised manuscript clarified my initial confusion. I have increased the score.

---

> > > ### Author Response · Authors · 2022-12-06
> > > **Thanks**
> > >
> > > Thank you for raising the score. We are sincerely grateful to your appreciation of our work.

---

### Official Review · Reviewer_tXY7 · 2022-10-29

**Confidence:** 4
**Correctness:** 3
**Technical Novelty And Significance:** 3
**Empirical Novelty And Significance:** 3
**Recommendation:** 5

**Clarity, Quality, Novelty And Reproducibility:**

The writeup is clear and the work has sufficient novelty. Code is included for reproducibility.

**Strength And Weaknesses:**

Strengths:
- Nicely written
- Baselines are chosen well
- Outperforms all baselines for both F1 and F2 (weaknesses highlight why this is not really valid)
- Evaluated on both SEEG and EEG datasets
- Includes clinical collaboration further signifying the significance of the work
- The domain adaptation experiment in Table 2 is also a great way to show the significance of the work


Weaknesses:
- Split table 3 into two tables?
- Domain adaptation not shown on TUSZ dataset
- Including the same subject in both testing and training invalidates the results

**Summary Of The Paper:**

The paper proposes an SSL framework for EEG and evaluate for seizure detection.

**Summary Of The Review:**

The paper is nicely written with good results on two different datasets. The demonstrated application is important and the strong performance indicates the significance of the work. However, including the same subjects in training and testing invalidates the results. It is not clear if that was done for other approaches as well. Even if it was, both should be done by selecting different subjects for training and testing.

---

> ### Author Response · Authors · 2022-11-17
> **Response to Weaknesses**
>
> **Weakness 1:** Thanks for your suggestion. We split Table 3 into two columns to align Table 1 & 2 and to save space. In the new version, we will move the table of the ablation experiment to the appendix and restore it to a single column table.
>
> **Weakness 2:**
>
> - The monitoring sites of EEG data are consistent among different patients, but the placement location and number of electrodes implanted in SEEG data vary greatly from patient to patient. **This results in a smaller patient distribution shift for EEG data than for SEEG data.** As a result, unlike EEG data, SEEG data needs to be fine-tuned on the target patient to achieve a good clinical performance. Following most of the existing work of epilepsy detection on EEG data (Asif et al., 2020; Covert et al., 2019), we also choose the setting that does not fine-tune the model on the target patient data.
> - In practical clinical applications, **SEEG data for each patient is usually recorded uninterrupted for days or even tens of days**. The long records contain dozens or hundreds of seizures, making fine-tuning the model with partial data a clinically acceptable option. Although TUSZ dataset contains a large number of patients, **there are not many seizures for each patient (approximately 10 seizure records per patient)**. If part of the data needs to be taken out for fine-tuning, the test samples for each patient will be further reduced and the test results will be more unstable.
> - In conclusion, we do not conduct domain adaptive experiments on the TUSZ dataset. But we are still grateful to your suggestion.
>
> Asif, Umar, et al. "SeizureNet: Multi-spectral deep feature learning for seizure type classification." *Machine Learning in Clinical Neuroimaging and Radiogenomics in Neuro-oncology*. Springer, Cham, 2020. 77-87.
>
> Covert, Ian C., et al. "Temporal graph convolutional networks for automatic seizure detection." *Machine Learning for Healthcare Conference*. PMLR, 2019.
>
> **Weakness 3:**
>
> We sincerely appreciate the suggestion. We have shown the new results in **Common Question 1**.

---

> ### Author Response · Authors · 2022-12-04
> **Look forward to your reply**
>
> Dear reviewer tXY7, so far, we have revised the article, added experiments or made clarifications through comments according to your suggestions. Do you have any further suggestions? We look forward to your reply and will follow up with your comments promptly.

---

### Author Response · Authors · 2022-11-17
**Response to Common Questions - Continued**

**Common Question 2 (More Ablation Experiments of MBrain)**

We have proposed two ideas on how to directly implement the multi-channel CPC in Section 2. For the second idea, we have reported the results of a shared CPC regarding all channels as one on the `CPC` row of Table 1. For the first idea, we design two strategies to combine multi-channel CNN or MLP into CPC respectively to learn representations for each channel.

- Directly use 1-Dimension CNN to encode the whole time series data and the number of channels during the process is $\mathbf{C} \rightarrow 256 \rightarrow 256 \rightarrow \mathbf{C} \times 256$, and split the output into $\mathbf{C}$ representations, each of which is a 256-dimensional representation. Then an LSTM is implemented to it. Then we execute the self-supervised task and the downstream task of CPC based on the representations for each channel as MBrain does, this variant is denoted as "CPC-Conv".
- As Reviewer Pj9B suggests, we use the contextual representations of all $n$ channels as input to an MLP in a fixed order, but we set the representation of the target channel to $0$ tensor when we aggregate them. By using the output of MLP as the aggregated representation of other channels, we perform subsequent experiments following exactly the same steps as MBrain. We name this variant as "CPC-MLP".

We would like to emphasize that both variants have more parameters than MBrain, while also ensuring that representations with the same dimension is learned for each channel for channel-wise epilepsy prediction on SEEG data. We report the results as below:

|      |  Models  |          Pre.           |          Rec.           |          $F_1$          |          $F_2$          |          AUROC          |
| :--: | :------: | :---------------------: | :---------------------: | :---------------------: | :---------------------: | :---------------------: |
| SEEG |   CPC    |     $27.65\pm4.49$      |     $55.07\pm3.52$      |     $34.20\pm3.40$      |     $42.73\pm2.57$      |                         |
|      | CPC-Conv | $6.39\mathbf{\pm0.77}$  |     $33.21\pm4.00$      | $10.53\mathbf{\pm1.07}$ | $17.46\mathbf{\pm1.42}$ |                         |
|      | CPC-MLP  |     $25.84\pm3.07$      |     $52.70\pm3.65$      |     $32.18\pm2.46$      |     $40.34\pm2.05$      |                         |
|      |  MBrain  | $\mathbf{37.97}\pm2.75$ | $\mathbf{65.07\pm2.68}$ | $\mathbf{46.45}\pm2.25$ | $\mathbf{55.28}\pm1.77$ |                         |
|      |          |                         |                         |                         |                         |                         |
| EEG  |   CPC    |     $22.81\pm2.04$      |     $58.31\pm7.55$      |     $32.50\pm1.24$      |     $44.02\pm2.43$      |     $74.53\pm1.00$      |
|      | CPC-Conv | $14.19\mathbf{\pm0.68}$ |     $54.22\pm4.51$      |     $22.46\pm0.89$      |     $34.58\pm1.59$      |     $61.07\pm1.20$      |
|      | CPC-MLP  | $\mathbf{22.87}\pm3.07$ | $64.92\mathbf{\pm3.49}$ | $33.80\mathbf{\pm0.59}$ |     $47.42\pm1.48$      |     $75.41\pm1.42$      |
|      |  MBrain  |     $22.13\pm1.03$      | $\mathbf{76.99}\pm4.49$ | $\mathbf{34.32}\pm0.90$ | $\mathbf{51.34\pm0.97}$ | $\mathbf{77.96\pm0.97}$ |

We can observe that the performance of "CPC-Conv" decreases dramatically. We speculate that this is because the channels are relatively independent, and the correlation between most channels is weak or even non-existent. Direct adoption of multi-channel convolution may introduce spurious and noisy correlations. However, the graph structure learning proposed by us has a sparsity assumption, and the representation extraction of each channel is relatively independent, so it can effectively learn and aggregate more significant information. For "CPC-MLP", we use an MLP to aggregate the representations of other channels, and then concatenate it with the representation of the target channel to predict future data. Unlike "CPC-Conv" which directly adopts multi-channel convolution for the raw data to obtain the "mixed" low-level representations, "CPC-MLP", like MBrain, learns the correlation of channels based on the "separate" high-level representations. Therefore, the performance of "CPC-MLP" does not drop as dramatically as that of "CPC-Conv". It can be observed that “CPC-MLP” outperforms "CPC" on the EEG dataset. This may result from the fact that the number of channels in EEG dataset is only 19, while that in SEEG dataset is 52 to 124. Consequently, the ablation results show that the graph structure learning we design has a reasonable and parameter-efficient inductive assumption.

---

> ### Comment · Reviewer_DmoA · 2022-11-20
> **Response to Common Question 2**
>
> - Are these results still follow the original split strategy? If so, I am afraid to disagree with this strategy. Please use subject-wise splits as you did for SEEG (e.g., 3-1-1).
> - To clarify. Using architecture choice, we can capture the correlation between channels, such as fusion strategy in the multi-modal setting. Another way to capture correlations is to have a coordinated setting when you encode each channel separately and then enforce correlation in the objective function. Using the graph network is a late fusion.
> - I would be careful to talk about spurious and noisy correlations because fusion models are still strong if both modalities/channels are available.
> - For architecture choices, it may be a good idea for authors to explore transformer architectures or MLP Mixer (Tolstikhin et al.). This is not the main reason to reduce the score, but a suggestion for future steps.
>   - Tolstikhin, Ilya O., et al. "Mlp-mixer: An all-mlp architecture for vision." Advances in Neural Information Processing Systems 34 (2021): 24261-24272.

---

> > ### Author Response · Authors · 2022-12-02
> > **Response**
> >
> > Thanks for your suggestion. We want to discuss the second point.
> >
> > - For the two variants we design in this table (CPC-Conv and CPC-MLP), they cannot process patients with different channels, because the input channel number of CNN and the input dimension of MLP are both predefined.
> > - Fusion strategy in the multi-modal setting aims to learn a joint representation that models cross-modal interactions between individual elements of different modalities, effectively reducing the number of separate representations. This type of approach fuses representations of all modes into a single representation and implicitly learns correlations and interactions between modes by optimizing target tasks. **In our scenario, since the model needs to make predictions for each channel/mode, it needs to learn different representations for each channel, rather than just obtaining one representation after all channels/modes are fused.** Specifically, for the task of multi-channel CPC, for each channel, we hope to use a method to aggregate the representation of correlated channels except for the target channel, so as to perform the prediction task. **Learning additive and multiplicative interactions between different modes through a fully connected network (e.g., CPC-MLP) is also one of the fusing methods.**
> > - Representation coordination aims to learn multimodal contextualized representations that are coordinated through their interconnections. In contrast to representation fusion, coordination keeps the same number of representations but improves multimodal contextualization. Representation coordination includes strong coordination and partial coordination. Strong coordination aims to bring semantically corresponding modalities close together in a coordinated space, thereby enforcing strong equivalence between modality elements. Partial coordination also enforce different types of constraints on the representation space beyond semantic similarity by partially coordinated models, and perhaps only on certain dimensions of the representation. **However, representation coordination cannot apply to our scenario because we we don't have explicit modal correspondence and semantic correlation, and coordinating representations between different modes is not a process of aggregating information.**
> > - Since we are not experts in the field of multimodal learning, we may miss the essence of it due to our superficial knowledge. We would appreciate it if you could talk about how multimodal learning can apply to our scenario more specifically.

---

### Author Response · Authors · 2022-11-17
**Response to Common Questions - Continued**

**Common Question 1 (Domain Generalization Experiment for SEEG data) - Continued**

- Due to the long record, clinical SEEG data contains tens or even hundreds of seizures, allowing the use of a subset of data to fine-tune the model and thus predict the rest of the data. Therefore, in order to emphasize **the clinical value of our model and reveal the potential value of correlation learning in detecting neurological diseases (Lee et al., 2022)**, we conduct supervised domain adaptation experiment. The experiment proves that despite the large distribution shift between different patients, the correlation learning between channels can still produce good initial parameters after being pre-trained, which reflects some common characteristics of epilepsy detection on SEEG. We believe that the experimental results have meaning of inspiration for the study of neurological diseases based on brain signals (especially on SEEG data).

Lee, Jin Hyung, Qin Liu, and Ehsan Dadgar-Kiani. "Solving brain circuit function and dysfunction with computational modeling and optogenetic fMRI." Science 378.6619 (2022): 493-499.

---

### Author Response · Authors · 2022-11-17
**Response to Common Questions**

**Common Question 1 (Domain Generalization Experiment for SEEG data)**

- As the EEG dataset contains a large number of subject data, experiments on EEG dataset are conducted with the setting splitting subjects as the training and testing sets. However, due to differences in the location and number of implanted electrodes, the distribution shift between patients in SEEG is much larger than that in EEG data. Not only our proposed model, but also all baselines are run under the same setting. To some extent, the results in Table 1 represent **an upper bound on the performance** of the current models. Only training and testing the models on the same patient can **achieve clinically competitive performance** with EEG data. The experimental results further indicate that **epilepsy detection with SEEG data is far more difficult than that with EEG**, which deserves further investigation.

- SEEG data is difficult to obtain, requiring a *craniotomy* to implant electrodes and record data for a long time. The SEEG data requires extensive procedures and approval before it can be used. The number of SEEG patients currently does not reach the size of EEG, and does not fully cover all types and brain regions of seizures. Besides, the SEEG dataset does not contain signals from normal subjects, as it would be extremely risky for a normal person to collect SEEG signals. Therefore, in the present stage, the results of domain generalization experiment on SEEG data will be more unstable.

- Nevertheless, we still report the performance of models under this setting. More specifically, we follow the `3-1-1` setting, where 3 patients are used for training, 1 patient is used for validation and 1 patient is used for testing. We conduct experiments for all combinations, pick up the best result for each patient, and report the average results over all patients.

  | Models |          Pre.           |          Rec.           |          $F_1$          |          $F_2$          |
  | :----: | :---------------------: | :---------------------: | :---------------------: | :---------------------: |
  | MiniRocket | $5.85\mathbf{\pm0.20}$ |     $\mathbf{39.18\pm0.59}$      |     $9.93\mathbf{\pm0.29}$  |   $17.24\mathbf{\pm0.37}$      |
  |  CPC   |     $22.88\pm5.06$      | $23.92\pm3.90$ | $20.11\pm3.27$ | $21.23\pm2.49$ |
  |  SimCLR   |     $14.02\pm3.71$      | $26.36\pm4.99$ | $11.07\pm3.49$ | $13.47\pm4.01$ |
  | T-Loss | $21.38\pm4.25$ |     $28.50\pm4.07$      |     $23.48\pm3.30$      |     $25.90\pm3.06$      |
  | TST | $8.37\pm3.96$ |     $32.48\pm8.25$      |     $11.80\pm3.91$      |     $15.67\pm3.69$      |
  | TS-TCC | $24.24\pm4.51$ |     $26.61\pm5.96$      |     $19.89\pm5.23$      |     $22.11\pm5.08$      |
  | TS2Vec |     $27.93\pm5.23$      |     $29.49\pm3.97$      |     $26.78\pm3.29$      |     $27.88\pm3.52$      |
  | MBrain | $\mathbf{30.69}\pm5.92$ | $38.94\pm4.34$ | $\mathbf{32.61}\pm3.60$ | $\mathbf{35.64}\pm3.04$ |

We point out that although GTS and MBrain are both graph-based SSL models, GTS cannot be trained on multiple patients, since the DCRNN encoder used in GTS can only process graphs with fixed nodes. In contrast, our model is designed to learn the correlations between each pair of nodes and utilizes inductive GNN, so it can easily handle graphs with different numbers of input nodes. In general, the performance of models under the domain generalization setting decreases significantly (40.37% on average in terms of $F_2$-score) compared with that in epilepsy detection experiment. The drop for recall metric is more pronounced, confirming that the distribution shift of patients in SEEG data is more significant than that in EEG. This results from the fact that different brain regions and different types of epileptic waves have different physiological properties and patterns. MBrain in this experiment still improves $F_1$-score and $F_2$-score by 21.77% and 27.83% respectively, compared to the best baseline. The results prove that MBrain has a superior generalization ability benefiting from rational inductive assumption of model design.

---

> ### Comment · Reviewer_DmoA · 2022-11-20
> **Response to Common Question 1**
>
> - Thank you for running subject-wise splits. It makes the model looks much better on SEEG.
> - It will be beneficial to discuss the invasiveness and difficulties of SEEG. Thus it might be essential to choose non-invasive EEG in the long run by showing that the EEG model can perform not worse or better than SEEG. It can be another important finding. In the end, we should balance with respect to trade-offs while being safety-critical in our choices.
> - I agree that you have more degrees of freedom in SEEG data, but we must keep evaluation strategies the same. It does not matter if there are some data collection reasons or data challenges. We want our models to generalize and understand underlying phenomena and not to learn some features that would work only on the same subject.

---

> > ### Author Response · Authors · 2022-12-02
> > **Response**
> >
> > Thanks for your meticulous reply and inspiring suggestions. We have learned a lot from all your comments. But we still want to argue some points as the following:
> >
> > - **Thank you for your suggestion on the analysis of the experimental results, but it goes against our original intention.** We agree that EEG data is easier to obtain than SEEG, which is riskier, and that EEG is a suitable and labor-saving method for researches that require healthy subjects. However, for many neurological diseases, neurosurgery operation has become a necessary means of recovery for patients because no specific mechanism has been found. In the case of epilepsy, for example, one-third of patients cannot control seizures with medication, most of whom give informed consent to a craniotomy to remove diseased brain tissue. On that basis, SEEG test is used to obtain more accurate information for the procedure. Because they don't want a lifetime of uncontrollable seizures compared to the risk of craniotomy. Therefore, SEEG cannot be replaced in clinical practice at this stage.
> >
> > - **We strongly agree that a really good model should learn very general features, which is also our ultimate vision.** But just as `Rome wasn't built in a day`, the design and exploration of our model will take a long time. Especially for hard-to-access SEEG data, the collection process is slow. Our current model is not as good at domain generalization because of the nature of the SEEG data, such as complex interaction patterns between large channels and physiological differences between patients. For another, the SEEG dataset does not yet cover all types and most brain regions. But we believe our model will get stronger as the number of patients we collect increases.
> >
> > - **Patient-specific models are not meaningless.** Because SEEG data is recorded over a long period of time, often containing dozens or hundreds of seizures, detecting seizure waves at this stage requires manual search from doctors, which is time-consuming and laborious. Our model is able to predict all of the remaining data of the target patient by finetuning with just a small part of the labeled data, which can significantly save time for doctors. **We recognize that a universal model which can completely take over the search process doctors do is indeed the simplest and most effective model, but the model for partial medical assistance does also have very practical applications.** Therefore, our original intention in designing the domain adaptation experiment is to demonstrate that our model can achieve the performance of the epilepsy detection task through a clinically acceptable scheme. The most important purpose of this experiment is to prove that **our proposed model has good clinical auxiliary function for doctors, which is also the basis for us to actually deploy the model into the auxiliary system**.
> >
> > - In conclusion, we totally agree with the value and necessity of designing a fully generalized model, which is why we supplement the results of the domain generalization experiment. But we cannot ignore the application value of auxiliary models, after all, for each specific patient, the shorter the duration of suffering from neurological disease, the better. We insistently argue that **our model is valuable as long as it improves the efficiency of clinical diagnosis and reduces the workload of doctors**. Therefore, the existence of epilepsy detection experiment and domain adaptation experiment are still necessary in our paper.

---

> > > ### Comment · Reviewer_DmoA · 2022-12-02
> > > **Response**
> > >
> > > I am willing to increase the score only if you provide additional EEG subject-wise experiments as you provided for SEEG. It will satisfy my empirical evidence criteria for any ML model. That is my primary concern. Note I asked about it from the beginning of the rebuttal period, and I expected that it wouldn't take too long to rerun the experiments with new splits.

---

> > > > ### Author Response · Authors · 2022-12-03
> > > > **Response**
> > > >
> > > > Thanks for your response and suggestion.
> > > >
> > > > - If we understand correctly, you are referring to a subject-wise experiment in which the training, validation and testing sets are divided by subjects. In fact, the experiment with EEG dataset has been conducted exactly in this way.
> > > > - Due to the large number of subjects included in the EEG dataset (more than 100 patients) but the very small number of seizures recorded for each subject (less than 10 seizures on average), we are unable to conduct the experiment on a single subject. The SEEG dataset contains a small number of subjects, but the amount of data for each subject is extremely large, so the experiment can be conducted on a single subject.
> > > > - We are sorry for the confusion. In order to clarify our settings more clearly, you can find a more detailed description of our settings (especially for the seizure detection experimental setting) in Section 4.2 of the new revision submitted on November 19th.
> > > > - We appreciate your suggestions to further improve the quality of our paper. Do you have any other questions or experiments that we need to supplement? We will answer and report relevant experimental results as soon as possible.

---

> > > > > ### Comment · Reviewer_DmoA · 2022-12-03
> > > > > **Response**
> > > > >
> > > > > If so, we need to improve clarity.
> > > > >
> > > > > Currently in Section 4.1:
> > > > > "For experimental efficiency, we generate a smaller
> > > > > dataset from TUSZ. We randomly sample 3,000 12-second EEG clips for self-supervised learning.
> > > > > As for the downstream seizure detection task, we first obtain 3,000 sampled 12-second EEG clips
> > > > > (80% for training and 20% for validation). For the testing, we sample another 3,900 12-second EEG
> > > > > clips with positive and negative sample ratio of 1:10."
> > > > >
> > > > > And later in Section 4.2.
> > > > > "For
> > > > > EEG data, there is no overlap between the patients of training and testing sets. "
> > > > >
> > > > > My understanding is that it is reasonable to consider the below.
> > > > > - We have $N$ subjects, first we split subject on $N_{CV}$ for cross-validation  and $N_T$ for hold-out test set.
> > > > > - Then cross-validation on $N_{CV}$ subjects is performed as K different stratified folds (based on coarse labels).
> > > > > - Given these splits, we sample for each subject $C$ clips.
> > > > > - Then, based on validation clips, we select checkpoints or tune hyperparameters.
> > > > > - Once we select checkpoint on validation, we perform the inference step for the $N_T$ test subjects and report the performance.
> > > > > - In the end, we never touched the hold-out test set for anything other than inference.
> > > > >
> > > > > Further, I understand that we perform adaptation for the test subject.
> > > > > - We split data for each test subject based on the clips.
> > > > > - We adapt based on training clips and select checkpoints and hyperparameters based on validation clips.
> > > > > - Then, we perform inference on the test clips.
> > > > >
> > > > > Ultimately, we should consider reporting numbers without adaptation, which is vital to understanding generalization. Further, it will be essential to understand how the generalization scales with respect to the number of subjects in the training data. Another option for evaluation is to transfer our model to a completely different dataset that has not been used for training.
> > > > >
> > > > > After this, we can perform an adaptation. Nevertheless, it will be beneficial to guarantee that adaptation does not lead to catastrophic forgetting (decrease in performance on other subjects). For example, we can perform linear evaluation (linear probing) with a frozen encoder and tune the classification head.

---

> > > > > > ### Author Response · Authors · 2022-12-04
> > > > > > **Response**
> > > > > >
> > > > > > Thank you very much for raising the score and logical sorting of our experimental results. According to your suggestion, we will further describe the specific steps of our experiments and the logic of the three experiments in detail.

---

### Author Response · Authors · 2022-11-19
**A new revision of our paper has been uploaded**

Dear reviewers, we have carefully and comprehensively revised our paper according to your comments. You can find corresponding reply on OpenReview of the major modifications of the paper, which have been highlighted in blue. We hope that our response and revision will address your main concerns. Could you please spare some time to review and evaluate our paper again? We are sincerely grateful to your valuable suggestions and will be glad to follow up with your further comments.

---

> ### Comment · Reviewer_DmoA · 2022-12-03
> **Recommendation adjustment**
>
> I want to thank the authors for their effort to address my concerns.
> I increased my score to "5: marginally below the acceptance threshold".

---

### Decision · Program_Chairs · 2023-01-20

**Decision:**

Reject

**Justification For Why Not Higher Score:**

The method is tightly coupled to neural data and rather complex. There is no core idea for ICLR readers to import into their own work. It is not clear if the method would have any utility for any other decoding tasks. This would be a great fit for a conference where seizure detection performance was the determining factor for publication.

Setting all of the above aside, questions about the methodology of the manuscript which are not fully resolved with reviewers make it impossible to recommend acceptance (specifically with reviewer DmoA, ending on Dec 4th with a pledge to clear up specific steps in the experiments performed). A pledge to provide further clarity on the experiments performed shows that the manuscript is not yet mature enough.

**Justification For Why Not Lower Score:**

N/A

**Metareview: Summary, Strengths And Weaknesses:**

Summary: Create embeddings of either EEG or SEEG data then use it to improve seizure detection.

Strengths: Investigating representation learning in other domains like neuroscience is of wide interest. The system appears to work better than baseline systems for SEEG and is marginally better at seizure detection with EEG data.

Weaknesses:
The method is tuned very specifically to neural data and is quite complex, making applications to other domains unlikely. There is no central key idea; the method essentially consists of three pieces that are combined together. This is fine, if the result is very flexible and generic; but the method is only demonstrated on seizure detection. Even there, improvements are only major on SEEG data, EEG shows a very incremental improvement.

Critically, the manuscript is unclear on specific details about experiments, for example, when it comes to adapting to multiple subjects. Authors promise to later improve the manuscript to clarify these points ("According to your suggestion, we will further describe the specific steps of our experiments and the logic of the three experiments in detail." on Dec 4th), but fundamental questions about what was done so late without an updated manuscript show that it is not yet ready for publication. At best the work is not reproducible in its current form, at worst, it was not fully described by the authors and thus parsed by the reviewers.

What a reader of the manuscript at ICLR could take away from this work is very unclear. As described in the manuscript itself, it is unclear if this method would improve performance on any other task. Had an array of decoding tasks been shown, readers at ICLR could imagine using this model.

The authors in their responses to reviewers to a range of concerns come back to one central point: the utility of this model to a clinical setting. To that end, a venue related to seizure detection would be a much better fit.